# A Comprehensive Review of Endogenous EEG-Based BCIs for Dynamic Device Control

**DOI:** 10.3390/s22155802

**Published:** 2022-08-03

**Authors:** Natasha Padfield, Kenneth Camilleri, Tracey Camilleri, Simon Fabri, Marvin Bugeja

**Affiliations:** 1Centre for Biomedical Cybernetics, University of Malta, MSD 2080 Msida, Malta; kenneth.camilleri@um.edu.mt; 2Department of Systems and Control Engineering, University of Malta, MSD 2080 Msida, Malta; tracey.camilleri@um.edu.mt (T.C.); simon.fabri@um.edu.mt (S.F.); marvin.bugeja@um.edu.mt (M.B.)

**Keywords:** brain–computer interface (BCI), brain–machine interface (BMI), electroencephalogram (EEG), endogenous, control, motor imagery (MI)

## Abstract

Electroencephalogram (EEG)-based brain–computer interfaces (BCIs) provide a novel approach for controlling external devices. BCI technologies can be important enabling technologies for people with severe mobility impairment. Endogenous paradigms, which depend on user-generated commands and do not need external stimuli, can provide intuitive control of external devices. This paper discusses BCIs to control various physical devices such as exoskeletons, wheelchairs, mobile robots, and robotic arms. These technologies must be able to navigate complex environments or execute fine motor movements. Brain control of these devices presents an intricate research problem that merges signal processing and classification techniques with control theory. In particular, obtaining strong classification performance for endogenous BCIs is challenging, and EEG decoder output signals can be unstable. These issues present myriad research questions that are discussed in this review paper. This review covers papers published until the end of 2021 that presented BCI-controlled dynamic devices. It discusses the devices controlled, EEG paradigms, shared control, stabilization of the EEG signal, traditional machine learning and deep learning techniques, and user experience. The paper concludes with a discussion of open questions and avenues for future work.

## 1. Introduction

This paper presents a systematic review of the literature related to online endogenous electroencephalogram (EEG) brain–computer interfaces (BCIs) for external device control. Controlling dynamic devices provides unique challenges, since they must be able to reliably perform actions such as navigating a cluttered space. Endogenous EEG control involves commands that are generated by actions executed by the users themselves and do not require external stimuli, as is the case for exogenous EEG control. An example of external stimuli is flickering lights, which are required for steady-state visually evoked potential (SSVEP) BCIs. 

This review aimed to fill a gap in the literature by focusing specifically on endogenous BCI control of dynamic devices. BCIs for a variety of devices, including exoskeletons, robotic arms, mobile robots, quadcopters, and wheelchairs, are discussed. This paper discusses issues that are particularly pertinent to the area of dynamic device control. It analyzes how commands are issued through different paradigms and the intuition of different approaches. It also discusses at length the issue of shared control, which involves merging data from sensors and commands issued through the BCI to ensure safer and easier operation. Other challenges addressed in this paper include the stabilization of the BCI decoder output to ensure reliable operation and methods for expanding the limited degrees of freedom that endogenous BCIs inherently have. In essence, this paper aimed to summarize the state of the art in these key areas, as well as highlight salient gaps in the literature. This review should give BCI researchers clear indications of options for future work within the area of dynamic device control. 

To increase objectivity, a systematic methodology was used to select which papers were included in the review. The paper opens with a summary of this methodology and then discusses the relevance of the review. It then briefly summarizes the proportion of papers featuring synchronous versus asynchronous BCIs in the literature. The main body of the review begins with a discussion of the endogenous paradigms used, the devices controlled through BCIs, and how commands are issued. Shared control methods are then discussed at length. The paper then reviews stabilization techniques for the BCI decoder output and methods used for overcoming the low number of degrees of freedom in endogenous BCIs. It also discusses novel techniques for handling errors in BCIs for dynamic devices through error-related potentials (ErrPs). Finally, the techniques used for EEG classification, which consist of traditional machine learning and deep learning techniques, are discussed, with a special focus on state-of-the-art techniques presented for dynamic device control. The paper then summarizes the number and type of subjects included in studies. This is an important issue because many BCIs for dynamic device control, such as BCI-controlled wheelchairs, are aimed at individuals with disabilities but are only tested with healthy subjects [1,2]. Testing on subjects with disabilities is also important because the low uptake of BCIs amongst target populations has been linked to a lack of understanding of the needs of possible end-users [3], and a fundamental step toward a better understanding involves including possible end-users in experiments [4]. Finally, this paper discusses the importance of user experience surveys and the results of these surveys that have been reported in the literature. If BCIs are to be widely adopted, it is important that researchers understand the subjective experience of users in order to determine functional areas that need improvement. This is especially important in BCIs for dynamic devices, since these can be used to navigate complex environments and execute physical tasks requiring precision, which may increase the mental load on the subject [5]. Results obtained through user-experience surveys could help improve the adoption rate of such technologies. The paper closes with a discussion of emerging questions and makes suggestions for future work. 

## 2. Methodology

The PRISMA methodology for selecting papers was used due to its focus on objectivity [6]. This methodology involves first establishing eligibility criteria, then conducting a database search, and then selecting papers through a screening process. 

### 2.1. Eligibility Criteria

In order to be included in this review, studies had to be published as journal papers. Journal papers were considered because they can be expected to contain significant research of high quality. Papers published through 31 December 2021 were considered, and retracted papers were excluded. No lower date limit was imposed on the dataset search. Studies had to test using human subjects and be written in English. The BCI presented had to include scalp EEG as a recording mechanism and be fully noninvasive, meaning any studies involving surgical implantation of electrodes were excluded. The papers had to present a novel BCI implementation and not just tutorials, editorials, or software/hardware technologies that could be applied to a BCI. Articles that were accepted but still in press were also reviewed. 

Only papers presenting purely endogenous BCIs were considered, thus studies which used both exogenous and endogenous brain signals were excluded. Exogenous signals are defined as any brain signals that require an external stimulus to be evoked, such as SSVEPs or auditory evoked potentials. Furthermore, the studies had to include an online component to the experimentation, in which brain signals were recorded and processed in real time. Since the review was focused on paradigms that can be used by subjects paralyzed from the neck down, papers that depended on limb-related motor execution paradigms were excluded. However, paradigms that involved facial movements such as blinks or jaw clenches were allowed. 

Papers had to present an active BCI, meaning that it enabled participants to control an external device. Passive BCIs designed purely for clinical monitoring or symptom detection in patients, such as those used for seizure detection or identification of depression, were not included. The BCI had to control a physical, dynamic device, such as an exoskeleton or wheelchair, and not just a graphical user interface application or virtual reality avatar. 

### 2.2. Information Sources and Search Strategy

This review was based on database searches conducted in PubMed, IEEE Xplore, and Scopus. These sources were chosen because of their strong links to engineering and science. The search terms used for each platform varied slightly due to their different interfaces, and were as follows:PubMed: (EEG) AND ((Brain-Computer Interface) OR (BCI)OR(Brain-Machine Interface)OR(BMI))AND((Online)OR(Real-Time)). This search was applied to the title/abstract and text. The following filters were applied to the search results to further refine the results: “Human Subjects” and “Journal Articles”.IEEE Xplore: (“Full Text & Metadata”:EEG AND (“Full Text & Metadata”:“Brain-Computer Interface” OR “Full Text & Metadata”:BCI OR “Full Text & Metadata”:“Brain-Machine Interface” OR “Full Text & Metadata”:BMI) AND (“Full Text & Metadata”:“Online” OR “Full Text & Metadata”:“Real-Time”)). The search was refined to “Journal Articles”.Scopus: (“EEG”) AND (“Brain-Computer Interface” OR BCI OR “Brain-Machine Interface” OR “BMI”) AND (“Online” OR “Real-Time”). The search was refined through the “Type” filter, which was chosen to be “Article”, and the “Source” filter, which was chosen to be “Journal”.

The final search was carried out on 14 June 2022.

Seven additional papers were considered based on recommendations from peers in the area of research on which this review paper focused. 

### 2.3. Selection Process

The search results from the databases were downloaded in spreadsheet format, and duplicate results were identified and removed. The first phase of selection was the abstract screening process, which involved reading the abstracts of each paper and classifying them as either ineligible for the review or possibly eligible. Once the abstract screening process was concluded, reading of the full text of the papers that were deemed possibly eligible was carried out, and these papers were marked as either eligible or ineligible for the review. Once this process was concluded, the eligible papers were considered for the review. 

Figure 1 shows a breakdown of the selection process. In total, 2557 unique search results were obtained. In the abstract screening process, 2047 records were excluded. The full texts of 510 papers were assessed, and papers were excluded for various reasons. Out-of-scope papers included studies based on exogenous systems, those designed for medical observation only, those that did not use any scalp EEG data, those related to limb-based motor execution, studies with nonhuman subjects, studies that were not about BCIs at all, and pieces that were not research articles. Of the papers recommended by peers, six were eligible for the study. Following this selection process, a total of 66 papers were included in the review. 

## 3. Relevance of this Review

In the database search and screening process, a total of 116 other review papers related to BCIs were identified. None of these were included in the 66 papers selected for this review because they did not present novel experimental BCIs. This section lists reviews that discussed EEG BCIs for dynamic device control, which is the main topic of this review paper. The relevance of this review within the context of these other review papers is then discussed. The relevant review papers were: Hekmatmanesh, 2021 [7]: Focus on brain-controlled vehicles, covering exogenous paradigms and endogenous paradigms.Wang, 2021 [8]: Review of BCI-controlled wheelchair systems, including electrode type, modality, and synchronicity.Wankhade, 2020 [9]: Focus on different EEG-based BCI paradigms, both exogenous and endogenous, and the signal-processing techniques used with them, as well as a brief discussion of online systems.Abiri, 2019 [10]: In-depth discussion of different EEG paradigms, including exogenous paradigms, with some discussion of signal-processing and classification techniques.Al-qaysi, 2018 [11]: Focused on EEG-based BCIs that drive wheelchairs, including exogenous and endogenous brain signals.

In the review paper by Wankhade et al. [9], the section discussing online BCIs was limited to three papers published in 2019, 2014, and 2010. The paper also had no in-depth discussion of methods of external device control. Our review paper focused on 66 papers presenting online BCIs for dynamic device control, and contains extensive discussion on how different endogenous BCI paradigms can be used to control external devices. 

The papers by Al-qaysi et al. [11] and Wang et al. [8] were focused only on BCI-controlled wheelchairs. The scope of our review paper was wider, with qualifying papers that presented any kind of BCI-controlled dynamic system being included in the review. Thus, BCIs related to exoskeletons, haptic robots, orthotics, prosthetics, robotic arms, pedaling machines, quadcopters, mobile robots, and wheelchairs are all discussed. This enabled our review paper to summarize the techniques used and overarching issues that exist across the entire range of BCI-controlled dynamic devices in the current literature. 

The review paper by Hekmatmanesh et al. [7] was largely focused on brain-controlled vehicles, whereas our review paper also discusses prosthetics, robotic arms, exoskeletons, and pedaling machines at length. The paper by Hekmatmanesh et al. [7] had a significant focus on the feature-extraction and classification techniques used, but some deep-learning-based systems were not discussed at length [2,12], whereas our paper discusses several deep-learning-based approaches in depth. Furthermore, our paper aimed to distill trends in the literature related to the mapping of mental activities onto commands, stabilization of the control signal, shared control (including a taxonomy), the issue of overcoming the low degrees of freedom in a BCI, and user experience. This level of detail on these issues was not present in the paper by Hekmatmanesh et al. [7]. Furthermore, the paper by Hekmatmanesh et al. [7] covered both exogenous and endogenous paradigms, but discussions of endogenous paradigms were largely limited to motor imagery (MI)-based and facial-expression paradigms, whereas our paper discusses other endogenous paradigms, including spelling-based and emotion-based paradigms. Moreover, several seminal works related to mobile robots that are mentioned in our paper [1,5,13,14] were not covered in Hekmatmanesh et al. [7].

The review by Al-qaysi et al. [11] mostly covered exogenous paradigms and only mentioned motor imagery (MI) as an endogenous paradigm in conjunction with an exogenous signal or another biosignal, such as an electromyogram (EMG). In comparison, our review paper presents purely endogenous paradigms that include traditional and sequential MI, facial movement, spelling- and emotion-based paradigms, and mixed paradigms that combined two or more of these methods. 

The review by Abiri et al. [10] had a strong focus on BCI paradigms, covering exogenous, MI, ErrP, and mixed paradigms. Although some of the endogenous paradigms mentioned by Abiri et al. [10] are also discussed in our review, our focus was not solely on the discussion of paradigms. Issues related to obtaining a stable control signal and shared control, as desired for the control of physical devices, were not discussed at length in the review by Abiri et al. [10], but they formed key discussions in our paper. Furthermore, the methods reviewed in Abiri et al. [10] were largely limited to conventional machine learning approaches, with just two mentions of neural or deep learning techniques. In our review, seven different neural and deep learning approaches are discussed. 

The review presented here is also more current, covering all papers through 2021. In contrast, the most recent review paper, by Wang et al. [8], did not reference any papers from 2021. The other review papers were published before 2021. 

Finally, the papers in our review were selected using a systematic approach that facilitated a comprehensive survey of the literature and reduced human bias in the selection of papers. The reviews of Hekmatmanesh et al. [7], Wang et al. [8], Wankhade et al. [9], and Abiri et al. [10] did not mention the use of this kind of systematic approach, which lent added value to our review paper. 

## 4. Overview of the Literature Growth across Time

This section presents a brief overview of how the literature we reviewed was spread out over the last two decades. Figure 2 contains a bar chart showing how many papers from each year were reviewed in the period from 2001, when the earliest paper was published [15], until 2021. The studies of dynamic device control that occurred during the early 2000s involved mobile robots [16,17] or orthotics/prosthetics [15,18,19]. The BCI competitions that took place between 2003 and 2008 [20,21,22,23] resulted in a substantial volume of BCI research in offline data throughout the mid-to-late 2000s [20,21,22,23]—this focus on offline processing may be one reason for the dearth of studies of dynamic device control from 2005 to 2011. Possibly boosted by this research into offline signal processing, the field of endogenous BCIs for dynamic device control began to pick up from 2012 onward, with a rapid growth in the literature occurring from 2012 until 2019. There was a sharp drop-off in publications in 2021, possibly due to the effects on research of the COVID-19 pandemic throughout 2020 and 2021. 

## 5. Synchronous vs. Asynchronous Control

The two main categories of device control were synchronous and asynchronous control. In synchronous control, the BCI indicates established intervals when the subject can issue a command. At the end of the interval, the data is processed and the command is executed. In an asynchronous system, the user can issue a command at any time. The BCI processes buffered input data and executes commands at regular intervals; for example, every 0.0625 s (a 16 Hz rate) [13]. Of the papers reviewed, 81% (54 papers) presented results that included an asynchronous control paradigm, while the rest presented results only for synchronous control paradigms. Although results generated using synchronous control could achieve high accuracies [24,25], they increase the latencies experienced by the subject and are not feasible for many practical BCIs; in particular, for the brain control of dynamic devices. This is because a synchronous control paradigm would lead to episodic movements in dynamic devices such as prosthetics, exoskeletons, and wheelchairs, which should ideally execute commands in real time to ensure smooth movement. 

## 6. BCIs in the Physical World: Applications and Paradigms

Figure 3 shows a breakdown of the different BCI-controlled dynamic devices in the reviewed papers. The most popular devices were mobile robots and assistive limb-movement devices (exoskeletons, orthotics, and haptic robots), with each having a share of 24%. The share for mobile robots went up to 38% if powered wheelchairs were included, as these are a form of specialized mobile robots. The share for assistive limb devices increased to 51% if prosthetics, robotic arms, and hands were included, representing articulated devices. The least common devices were unmanned aerial vehicles (UAVs), prosthetics, and pedaling machines. Most of the UAVs were quadcopters [26,27,28,29], with only one study presenting another type of UAV [30]. UAVs present numerous challenges in terms of their many degrees of freedom and difficulty in mastering control, even when using a manual remote controller. Pedaling machines are typically used in stroke rehabilitation studies [31], which constitute a smaller niche of research. Finally, prosthetics are generally aimed at subjects who have amputations but generally have reasonable muscle signals in the residual (stump) limb, which has led to the development of numerous reliable EMG devices [32]. These constitute a viable alternative to BCI-controlled prosthetics. Conversely, BCI-controlled exoskeletons, orthotics, mobile robots, wheelchairs, and robotic arms can all be used by subjects who have spinal or neurological injuries resulting in reduced or negligible residual muscle signals in limbs, creating research openings for BCI control of such devices.

There is a mutual relationship between the application for which a BCI is designed and the endogenous paradigm used. Table 1 summarizes the applications, paradigms, number of classes, and control functions for EEG-controlled devices in papers published from 2017 through the end of 2021 that represented the state of the art. In the table, the number of classes is the effective number of classes associated with device control. The table is organized first according to the EEG paradigm used, then by device, and then by year. Table 1 shows details for systems that used a single EEG paradigm, such as a traditional MI. Table 1 also includes a column for the “Accuracy” (also called success rate in some papers [1]) values quoted in studies. Although these results can give an idea of performance, different studies had different experimental setups; for example, some studies of BCI-controlled mobile robots included obstacles that they needed to avoid [12], whereas others did not [13]. Not all studies quoted an accuracy or success rate statistic, and used other performance measures such as sensitivity and specificity instead [33]. These papers are listed with ‘N/A’ in the Accuracy column. Table 2 contains similar details of systems that used mixed paradigms that combined more than one paradigm. 

### 6.1. Motor Imagery Paradigms

Considering Table 1, traditional motor imagery, which consists of MI related to distinct limbs, such as movement in the hands, feet (or legs), and tongue, was the most popular paradigm. This paradigm presented a particularly intuitive method of control for exoskeletons [34,35], in which the user can imagine walking or sitting [35]; rehabilitative pedaling machines, in which the user can imagine the pedaling motion [36]; and robotic hands, in which the hand can mimic imagined flexion or extension [37,38]. Many systems depended wholly or in part on classifying left- and right-hand MI, possibly because these produced distinct pattens in the right and left hemispheres, respectively, that can be accurately classified [15]. Furthermore, these MI actions can be intuitively related to left and right steering commands for mobile devices such as robots. Nevertheless, systems based on traditional MI tended to use four commands or less, possibly because it was challenging to accurately decode multiclass MI [28,29]. Furthermore, MI-based paradigms required significant user training for reliable use, and could suffer from a relatively high BCI illiteracy rate [4]. Enhanced intuition and control of MI-driven prosthetics and orthotics could be achieved by adding functional electrical stimulation, which activated the subjects’ own muscles [39,40].

In 2018, Yu et al. [1] presented a sequential motor imagery and finite-state-machine approach for controlling a motorized wheelchair. The sequential motor imagery approach involved the subject issuing commands by completing two mental activities sequentially (for example: left-hand and then right-hand MI), as opposed to just one mental command (for example: left-hand MI), as in the traditional MI paradigm. The finite-state machine is used to enable the same sequential command to be used for different actions, depending on the current state of the system. These approaches were used to increase the degrees of freedom of the BCI, and are discussed in more depth in Section 9. 

Jeong et al. [41] presented a BCI for controlling a robotic arm based on single-limb MI. Decoding single-limb movements was more challenging than decoding the movements of different limbs, as in traditional MI [41]. However, it presented an opportunity for more intuitive control paradigms, with the user being able to control the movements on a robotic arm in 3D space by imagining similar movements in their own limb. 

In order to master suitable control of MI-based BCIs, subjects require extensive training [4]. Traditional MI EEG subject training schemes used the discrete trial (DT) approach, in which subjects were directed by cues to execute a MI task for a certain amount of time and were given on-screen feedback; for example, through a progress bar or a moving object [4,42]. Edelman et al. [42] presented a novel training method called continuous pursuit (CP), in which subjects were trained through a gamelike interface that required them to control a cursor in following a randomly moving icon on-screen. They noted that subjects were more engaged, elicited stronger brain signal patterns, and could potentially be trained in less time when using CP as opposed to DT. In test trials involving a robotic arm, the level of performance obtained during CP training for cursor control was maintained, indicating that the CP approach was effective for training subjects to control a dynamic device. 

**Table 1 sensors-22-05802-t001:** A summary of EEG paradigms, devices, and control functions in the literature for single-paradigm systems.

Paper	Paradigm	Device	No. of Classes	Classes and Control Function	Accuracy
Choi, 2020 [35]	Traditional MI	Lower limb exoskeleton	3	Gait MI—walking; sitting MI—sitting down; idle state—no action	86%
Gordleeva, 2020 [43]			2	MI of dominant foot—walking; idle—standing still	78%
Wang, 2018 [44]			3	Left-hand MI—sitting; right-hand MI—standing up; feet MI—walking	>70%
Liu, 2017 [45]			2	Left-hand MI—moving left leg; right-hand MI—moving right leg	>70%
Ang, 2017 [46]		Haptic robot	2	MI in the stroke-affected hand; idle state	~74%
Cantillo-Negrete, 2018 [47]		Orthotic hand	2	MI in dominant hand (healthy subjects) or stroke-affected hand (patients)—moving; idle state—do nothing	>60%
Xu, 2020 [48]		Robotic arm	4	Left-hand MI—turn left; right-hand MI—turn right; both hands—move up; relaxed hands—move down	78% (for left vs. right and up vs. down experiments); 66% (for left, right, up, and down experiments)
Zhang, 2019 [49]			3	Left-hand MI—turn left; right-hand MI—turn right; tongue MI—move forward	73%
Xu, 2019 [50]			2	Left-hand MI—left planar movements; right-hand MI—right planar movements	>70%
Edelman, 2019 [42]		Robotic hand	4	Left-hand MI—left planar movements; right-hand MI—right planar movements; both-hands MI—upward planar movements; rest—downward planar movements	N/A
Spychala, 2020 [37]			3	MI hand flexion or extension for similar behavior in robotic hand, idle state—maintain hand posture	~60%
Moldoveanu, 2019 [51]		Robotic glove	2	Left-hand MI and right-hand MI—controlled movement of robotic glove	N/A
Zhuang, 2021 [52]		Mobile robot	4	Left MI—turn left; right MI—turn right; push MI—accelerate; pull MI—decelerate	N/A (>80% for offline)
Batres-Mendoza, 2021 [12]			3	Left-hand MI—turn left; right-hand MI—turn right; idle state—maintain behavior	98%
Tonin, 2019 [13]			2	Left-hand MI—turn left; right-hand MI—turn right. Idle rest state inferred from probability output of classifier.	~80%
Hasbulah, [53], 2019			4	Left-hand MI—turn left; right-hand MI—turn right; left-foot movement—move forward; right-foot movement—move backward	64%
Ai, 2019 [54]			4	Left-hand MI—turn left; right-hand MI—turn right; both-feet MI—move forward; tongue MI—move backward	80%
Jafarifarmand, 2019 [55]			2	Left-hand MI—turn left; right-hand MI—turn right	N/A
Andreu-Perez, 2018 [14]			2	Left-hand MI—turn right; right-hand MI—turn left. If the probability output of the classifier was less than 80%, maintain current state.	86%
Cardoso, 2021 [31]		Pedaling machine	2	Pedaling MI—cycle; idle state—remain stationary	N/A
Romero-Laiseca, 2020 [56]			2	Pedaling MI—cycle; idle state—remain stationary	~100% (healthy subjects); ~41.2–91.67% (stroke patients)
Gao, 2021 [40]		Prosthetic leg	3	Left-hand MI—walking on terrain; right-hand MI—ascending stairs; foot MI—descend stairs	N/A
Yu, 2018 [1]	Sequential MI	Wheelchair	6	Left hand, right hand, and idle state identified by classifier. Four commands obtained by sequential paradigm, used to execute six functions through a finite-state machine: start, stop, accelerate, decelerate, turn left, turn right.	94%
Jeong, 2020 [41]	Single-limb MI	Robotic arm	6	MI of same arm moving up, down, left, right, backward and forward, which were imitated by the robotic arm.	66% (for a reach-and-grab task); 47% (for a beverage-drinking task)
Junwei, 2018 [25]	Spelling	Wheelchair	4	Spell the desired commands: FORWARD, BACKWARD, LEFT, RIGHT	93%
Kobayashi, 2018 [57]	Self-induced emotive State	Wheelchair	4	Delight—move forward; anger—turn left; sorrow—turn right; pleasure—move backward.	N/A
Ji, 2021 [58]	Facial movement	Robotic arm	3	Detect double blink, long blink, and normal blink (idle state) to navigate VR menus and interfaces to control a robotic arm	N/A
Li, 2018 [59]		Prosthetic hand	3	Raised brow—hand opened; furrowed brow—hand closed; right smirk—rightward wrist rotation; left smirk—leftward wrist rotation	81%
Banach, 2021 [24]	Sequential facial movement	Wheelchair	7	Eyes-open and eyes-closed states identified by the classifier. Seven commands generated using three-component encodings of the states. Commands: turn left, turn right, turn left 45°, accelerate, decelerate, forward, backward.	N/A
Alhakeem, 2020 [60]			6	Eye blinks and jaw clench were used to create six commands using three-component encodings: forward, backward, stop, left, right, keep moving.	70%

### 6.2. Spelling and Induced Emotions

Other endogenous paradigms included spelling [25] and self-induced emotion paradigms [57]. Spelling [25] is a control paradigm that could present an intuitive control approach. However, only one system in the literature used this method, applying it to wheelchair control [25]. Furthermore, although the system obtained a success rate in trials of over 90%, it was a synchronous system. The success rate denoted the rate of correct completion of a navigation task on the wheelchair. Moreover, the idle state was not included as a class; thus, to keep moving forward, the user had to continuously spell the word “FORWARD”, which was mentally taxing and impractical. Self-induced emotions [57], which can be generated by recalling memories or emotive concepts, were found to be effective for asynchronous control of a wheelchair, with a success rate between 73% and 85% in executing four commands. However, induced emotions are a highly counter-intuitive approach because emotions cannot be logically related to navigational commands; for example, induced anger and sorrow were used to issue commands to turn left or right [57]. This was in contrast to more intuitive systems based on spelled commands such as “LEFT” and “RIGHT” [25] or imagined left-hand and right-hand movements [55] that could be easily associated with the commands to turn left and right. Furthermore, users had to make an effort to self-induce emotions despite their current mood, which may be challenging in everyday life. Self-inducing emotion may also be more mentally demanding than other paradigms, such as facial movement, which use relatively straightforward actions such as blinks [58,59]. 

### 6.3. Facial-Movement Paradigms

Facial-movement paradigms have been touted as a viable alternative to other endogenous paradigms such as MI because they have much lower BCI illiteracy rates [59]. This is because facial movements such as blinks and eyebrow or mouth movements produce distinct signal characteristics in EEG data that can be detected with a relatively high accuracy [58,59]. Sequential facial-movement paradigms have also been used to augment the number of commands executed using a small number of facial actions. These operate in a similar way to sequential MI commands, requiring the user to carry out sequences of facial actions, typically three [24], to generate commands. As with sequential MI commands, this requires the user to remember sets of arbitrary sequences of commands, which may require training for successful adaptation. Facial-movement paradigms can lead to counterintuitive or impractical commands such as a furrowed brow for closing a prosthetic hand [59] or closing one’s eyes for prolonged periods of time while navigating a wheelchair [24]. 

### 6.4. Multiparadigm Systems

Table 2 shows the details of mixed-paradigm systems. Three of these systems merged traditional MI with other paradigms to attempt to overcome the relatively poor classification accuracy of MI systems when compared to exogenous systems such as SSVEP [9]. 

Ortiz et al. [61] proposed a mixed paradigm based on MI classification and attention detection for controlling an exoskeleton. Attention detection is used to identify when the subject has a high attention level, which can occur when concentrating on generating MI. They applied this system to a two-class problem: MI for walking, which made the exoskeleton walk; and the idle state, which led to stopping. In this approach, the EEG data was passed through two independent decoders, one to identify the MI and one to identify the attention level. They then merged the decisions of the two decoders in an ensemble based on the certainties of each decoder output. Using MI and attention together resulted in an accuracy of 67%, compared to 56% for MI and 58% for attention alone. In the online testing phase, as given in Table 2, the traditional MI paradigm outperformed the MI and attention paradigms, and the authors hypothesized that this was due to lags in the online system. In all the experiments presented, the accuracies were relatively low, considering it is widely accepted that having an accuracy of over 70% is required for a reliable online system [4]. 

**Table 2 sensors-22-05802-t002:** A summary of EEG paradigms, devices, and control functions in the literature for multiparadigm systems.

Paper	Paradigm	Device	No. of Classes	Classes and Control Function	Accuracy
Ortiz, 2020 [61]	Traditional MI + attention	Lower-limb exoskeleton	2	Walk MI—walking; idle—just stand	Traditional MI: 63%; MI + attention: 45%
Tang, 2020 [2]	Traditional MI + facial movement	Wheelchair	4	Left-hand MI—turn left; right-hand MI—turn right; eye blink—go straight	84%
Kucukyildiz, 2017 [33]	Mental arithmetic + reading	Wheelchair	3	Idle—turn left; mental arithmetic—turn right	N/A

Tang et al. [2] merged a facial-expression paradigm and motor-imagery paradigm to expand the number of commands. A wheelchair could be controlled through left-hand MI, right-hand MI, or blinking to turn left, turn right, or move forward, respectively. However, this paradigm required continuous deliberate blinking to move forward, which may be uncomfortable. 

## 7. Shared Control

Shared control involves merging the BCI decoder output with information from environmental sensors to control the dynamic device. Shared control has been mainly used to implement emergency stopping, obstacle avoidance, and semiautonomous navigation in dynamic devices. 

A total of 27% of the papers reviewed (18 papers) explicitly discussed shared control as a part of the proposed systems. However, this did not mean that the other systems did not use information from sensors on the dynamic devices to regulate the behavior of the system. In fact, it was common for the BCI to be used to issue high-level commands to devices such as exoskeletons, orthotics, or robotic arms with grasping capabilities, then these devices executed those commands using internal sensors for guidance [40,43,44,62,63,64,65]. This section, however, reviews papers that explicitly discussed shared-control strategies as part of the presented work. 

Shared control presents a trade-off between the autonomy of the user in controlling the system through the BCI decoder and the control actions taken based on feedback from sensors. This leads to different levels of shared control, with higher levels of shared control leading to a greater impact of the sensor information on device performance. This section discusses these levels of shared control, which ranged from passive communication of sensor information to the user at the lowest level to semiautonomous systems that could navigate to a target selected by the user at the highest level. 

The lowest level of shared control in the reviewed literature facilitated passive communication from the dynamic device to the user [66]. The earliest examples of this were in the BCI-controlled quadcopter studies by LaFleur et al. [27] (2013) and Kim et al. [26] (2014), in which a video stream recorded from the hull of a quadcopter was fed back to the subject. Later, in the study by Li et al. [66], an RGB sensor was attached to a mobile robot, and incoming images were processed using a deep learning simultaneous localization and mapping (SLAM) algorithm to identify objects. The maps were then displayed on a graphical user interface (GUI) with possible obstacles highlighted. The user was responsible for carrying out obstacle avoidance based on this feedback. This type of shared control would be useful for applications involving remote control, such as robots that explore debris after an earthquake or flying a drone from another location. This is the lowest level of shared control, since it merely feeds back sensor information to the user, leaving them to carry out obstacle avoidance and navigation. 

Gandhi et al. [67] presented a slightly higher level of shared control in which the user could control the movements of a mobile robot by selecting directional commands such as left, right, forward, backward, and halt from an intelligent adaptive user interface. The intelligent adaptive user interface ordered the commands according to sensor feedback so that the commands facilitating obstacle avoidance were at the top of the menu, whereas more risky commands were toward the bottom of the menu. This made it easier for the subject to easily select the recommended commands that would support obstacle avoidance while still giving them full autonomy to select any navigation command in the system. Menu navigation was carried out through MI commands. 

A similar approach was presented by Shi et al. [30] for an UAV. A laser range finder and video camera were used to identify obstacles in the vehicle’s path. When an obstacle was detected, the UAV hovered and the user was presented with a menu of options for commands that would facilitate obstacle avoidance (the options could consist of turn left, turn right, or continue ahead, depending on the situation). The user could, however, choose to not take the suggested actions, override the menu, and then control the UAV fully through the BCI. 

Some studies used shared control for emergency stopping. Emergency-stopping systems offer a higher level of shared control, since the feedback from the sensors can automatically stop movement of the device, thus superseding the BCI decoder output during emergency situations. Infrared [24], ultrasound [12], and Kinect sensors [33] have all been used to detect obstacles and stop the movement of a mobile device if an obstacle was nearby. Once the device was stopped, the user was required to use BCI commands to navigate around the object. 

Navigating around obstacles manually through a BCI, however, can be tedious [5]. Furthermore, some devices, such as robotic manipulators, require fine movement control that may not be possible with current EEG-based BCI decoders [68]. These issues can be addressed through higher-level shared control systems that implement automatic obstacle avoidance or the execution of fine motor control. Obstacle-avoidance algorithms use incoming sensor data to guide the movement of a device around objects. The earliest example of obstacle avoidance being used with a BCI for dynamic device control was in the 2003 work of Millán et al. [16]. In this BCI, the subject controlled the direction of travel (left, right, or forward) of a mobile robot through a maze, and the mobile robot could perform obstacle avoidance or emergency stops automatically if there was a risk of a collision. When the robot encountered a wall, it followed the wall until the subject issued a command that moved the robot away from the wall. Details of the control algorithms were not included in the paper. 

A later study [52] implemented obstacle avoidance such that whenever an object was identified, the mobile robot maximized the distance between the robot and the obstacle. During obstacle avoidance, the BCI-issued user commands were ignored. Although this approach prevented collisions, maximum margin obstacle avoidance may be unsuitable for applications in which subjects need to approach objects at close range; for example, to inspect them or pick them up.

In 2015, Leeb et al. [5] presented a telepresence robot that used the artificial potential fields method to enable the robot to approach objects at close range. Artificial potential fields are a standard path-planning approach [5,69] and involve modeling objects in the environment as either attractive forces, which pull the planned path toward them, or repellant forces, which push the path away. Infrared sensors on the robot were used to identify nearby objects [5], and the objects were modeled by attractive forces if the BCI decoder indicated that the direction of travel was toward that object. Otherwise, the object was modeled by a repulsive force. Using this approach, the telepresence robot could navigate a cluttered living environment. The shared control approach was effective: when comparing the average results obtained when using shared control to when the robot was controlled with just the BCI decoder; using shared control resulted in trials that were 33% faster and required 15 fewer commands. 

Another study used artificial potential fields to guide the movement of a robotic arm toward targets based on the BCI decoder output and targets identified through a Kinect sensor [69]. In this work, the objects were detected using the Kinect sensor, and the BMI decoder’s output gave an indication toward which target the subject would like to move the robotic arm. This target was then assigned an attractive force, and the arm moved toward the target. A blending parameter was used to balance the influence of the BCI decoder and the sensor on the movement of the arm to ensure that the user could exert influence over the system. This blending parameter was used to fuse the robotic arm velocity obtained from the BCI decoder with the ideal velocity vector, which would transport the arm directly toward the target that the system predicted the subject wanted to reach. This blending parameter ensured that if the system had incorrectly interpreted the direction of movement desired by the subject, the arm did not immediately move to the incorrect target, while at the same time it enabled the sensor data to aid the movement of the arm toward the desired object, possibly reducing mental fatigue and frustration in the user [69]. 

Shared control has also been used to facilitate finer reach and grasp motions in a robotic arm, which can be challenging to execute when using just the BCI decoder output [49,68]. For example, Li et al. [68] facilitated control of a robotic manipulator in three dimensions through a shared-control strategy. They divided the workspace of the manipulator into two areas: an inner area where the targets were located, and an outer area. The manipulator was governed by different control laws depending on the area. When in the outer area, the user commands issued by the BCI for moving left, right, up, down, forward, and backward were closely followed; however, when the manipulator joint was pushed outside of the bounded outer area, the joint was reset, pulling the manipulator toward the inner area again. The inner workspace was divided into a fine grid of possible locations, and the navigation of the manipulator was guided toward one of these definite locations. At each update, the manipulator was moved to one of the six grid-points neighboring its current position, and the location selected depended on the current location of the manipulator and the output command from the BCI. This enabled finer control of the movement of the manipulator within the inner area. 

Shared control has also been used to implement clean-cut control modes between the BCI and the underlying controller [50]. In the approach by Xu et al. [50], the user, through the BCI decoder, was generally in full control of the movements of a robotic arm. However, the arm was equipped with a depth camera that could sense targets, and once the arm was within a certain distance from a detected target, an autonomous controller took over the grasping motion to seize the target object. This removed the need for a user to issue a grasping command, and saved the user from the possibly frustrating process of eliciting fine motor control through the BCI alone. However, this approach may be inappropriate for grasping objects in a crowded environment, since the object on which the subject is focused may not be clear. 

Semiautonomous systems provided the highest level of shared control in the reviewed literature. The gaze-and-BCI upper limb exoskeleton presented by Frisoli et al. [70] enabled a subject to select a target object through eye-gaze tracking and activate the upper limb exoskeleton through MI, and then the exoskeleton moved autonomously to grasp the target. A Kinect camera was used to identify the objects in the subjects’ view range, and then the eye-gaze tracking data was transposed onto the segmented Kinect image in order to identify the object at which the subject was looking. Another semiautonomous system was the BCI-controlled wheelchair by Zhang et al. [71], in which the subject selected a destination from a menu, and then the wheelchair performed path planning, path following, and obstacle avoidance to guide the subject to the desired destination. Menu navigation was carried out using an MI paradigm, while path planning was carried out using the popular A* algorithm, which finds the shortest path between two locations, and proportional–integral control was used for path following [71]. 

Figure 4 shows a taxonomy for the shared control approaches discussed throughout this section. Based on the reviewed literature, shared control could be split into two major branches: those that just passed sensor feedback to the BCI user [66,67], and those that had some element of automation involving a controller that executed actions based on sensor data. Shared-control paradigms with an element of automation could be further divided into those in which the BCI always had influence over the behavior of the system, even when obstacle avoidance was being carried out [5,69], and those in which the input from the BCI was sometimes ignored, with the sensor-based controller completely taking over operation at certain times. The latter type of shared control could be clearly grouped into two branches: the first consisted of BCIs that enabled users to issue low-level commands and an automatic controller with an assistive nature that was designed to take over operation infrequently during specific events such as emergency stops [12,24,33,50,68]; the second consisted of semiautonomous systems in which the BCI was used to issue high-level commands, such as selecting the destination for a BCI-controlled wheelchair, and then the sensor-based controller executed those commands [70,71]. 

Although systems with higher levels of shared control may reduce the mental workload on the subject by enabling them to navigate complex tasks faster and with fewer commands [5], as the level of shared control increases, care must be taken to ensure that the will of the user is not overridden by the sensor-based control algorithm. Since the desired level of shared control may vary between subjects, tunable systems that allow for variation in the level of shared control, such as in the work of Kim et al. [69], may be important for commercial BCIs. 

## 8. Obtaining Stable Control from BCI Decoders

The majority of the papers we reviewed obtained the control signal directly from the output of the BCI classifier [2,28,31,33,37,43,46,48,50,56,59,60,61,63,66,72]. For asynchronous systems, this generally involved windowing the EEG data and obtaining a classification output at a regular rate [2,28,33,37,43,46,48,50,56,59,60,61,63,66,71,72]. This classification output was used to drive the external device. However, BCI decoders are known to have an unstable output that is prone to spurious misclassifications [13,54], and some of the reviewed papers attempted to mitigate this to provide a more stable BCI control signal for control [13,35,54]. The two main approaches in the literature for obtaining stable control signals were false-alarm approaches and smoothing approaches. 

### 8.1. False-Alarm Approaches

The false-alarm approach was the most straightforward one used in the literature to obtain a smoother control signal [52,54]. This approach is based on the assumption that a subject cannot change their mental state instantaneously. Thus, when the system identifies a change in mental state at the classifier output, either dynamic operation is paused until the output stabilizes, or the current state is maintained until a certain number of consecutive classifier outputs are of the same state. 

In 2012, Chae et al. [73] presented a comprehensive fading feedback rule that was based on the false-alarm approach. In this approach, in order to execute a command, a “buffer” of four similar consecutive classification outputs for the command had to be achieved. Once the buffer was full, the system continued to execute the command as long as the BCI output remained consistent. If the classification output did not match the current control state, command execution paused, and one level was removed from the buffer. If the subject wished to continue in the current command state, then they had to issue a correct classifier output to top up the buffer before execution of the command could recommence. Otherwise, to change state, the subject had to execute three more BCI decoder outputs that were not the current state to reduce the buffered values to zero, and then execute four consecutive correct classifications of the new desired state to refill the buffer and begin executing the new command. This meant that the fastest way to change state involved the BCI classifier outputting the same label for eight consecutive samples. Chae et al. [73] verified that this approach improved the classification accuracy, but resulted in longer decision times for the BCI system.

In 2015, Hortal et al. [74] presented a slightly different approach for controlling a robotic arm: the latest five consecutive classification labels determined by the BCI decoder were considered, and if all five were the same the movement, the command was executed; otherwise, no action was taken and the arm remained at rest. 

More recent approaches, such as those of Ai et al. [54] (2019) and Zhuang et al. [52] (2021), used more straightforward approaches for false-alarm systems that only required two or three consecutive samples, respectively, to be similar. This introduced lower latency than the eight-step process for changing the buffer state in the work of Chae et al. [73]. Although Ai et al. [54] commented that the false-alarm system could reduce false positives, this benefit was not verified experimentally in [54] or by Zhuang et al. [52], and neither cited literature to support this design choice. Verification of the effectiveness of the false-alarm approach is desirable. 

Figure 5 compares the operation of the four different false-alarm approaches discussed in this section. This illustrative example is for a two-class problem, in which classes A and B are related to different mental states that cause different types of movement in the dynamic device. The blue bars show the output of the BCI classifier at the previous time step, and the orange bars show the action taken in the dynamic device at the current time step. It is immediately evident that there was a significant philosophical shift in these false-alarm approaches: the earlier approaches of Chae et al. [73] and Hortal et al. [74] halted the operation of the device when the BCI classifier outputted a different label and a false alarm was triggered, while the more recent approaches by Ai et al. [54] and Zhuang et al. [52] just maintained the current device state through the false-alarm condition. By halting operation earlier, approaches could successfully reduce the time spent executing an unwanted command, but this resulted in episodic movements of the device, which could affect the smoothness of operation that subjects might expect in an asynchronous BCI. Later works that maintained the current state gave the user a better impression of continuous device control when spurious misclassifications occurred. However, they could remain in a control state for longer than the subject intended if the classifier was susceptible to high levels of instability. 

### 8.2. Smoothing Approaches

Some works in the literature smoothed the control signal [5,13,41,45]. The most straightforward approach in the literature involved averaging the label probabilities outputted from the classifier over a number of consecutive time steps, with studies using two [41], three [45], four [75] or eight steps [16]. Leeb et al. [5] used a weighted average of the present BCI decoder output and the previous smoothed output sample. The weighting parameter could be tuned for each individual subject. The class associated with the highest smoothed probability was the class assigned in the control signal. 

In 2019, Tonin et al. [13] presented a dynamical smoothing method that drove a control signal to one of three stable points, corresponding to left- and right-turn actions for mobile robot control and the idle state, which corresponded to no robot-turning action while the robot continually moved forward. The approach involved calculating two mathematical forces that were then used to determine the change in the control signal from one time step to the next. The first force was influenced by the value of the control signal at the previous time step, while the second was influenced by the BCI decoder output at the current time step. The proposed approach was compared to the weighted average smoothing approach of Leeb et al. [5] for navigating a mobile robot toward one of six targets within a fixed space. The approach by Tonin et al. resulted in a shorter length of the trajectory to reach a predefined target and a higher accuracy in terms of times the targets were successfully reached. 

## 9. Overcoming the Limited Degrees of Freedom in Endogenous BCIs

One of the core issues in endogenous BCIs is that even with state-of-the-art signal-processing and classification techniques, the number of classes that can be accurately classified is limited [1,28]. This is evident in Table 1, which shows that endogenous systems based on mental commands were often limited to four commands or less. Since dynamic devices typically have a high number of degrees of freedom, innovative solutions have been found to facilitate their control through endogenous BCIs. The main solutions in the reviewed literature were sequential control paradigms, hybridization of the BCI, and menus that could be navigated through limited commands. 

### 9.1. Sequential Command Paradigms

Sequential command paradigms require the subject to issue commands by executing two endogenous mental activities sequentially, as briefly introduced in Section 6. A sequential paradigm involves training a classifier on a number of singular endogenous activities, such as left-hand MI. However, during online use of the BCI, subjects issue commands by sequentially carrying out mental activities; for example, imagining left-hand MI followed by right-hand MI. Postprocessing of the classifier output is used to identify the sequential activities and thus the command to be issued. 

The earliest reporting of a sequential command paradigm was in the 2014 synchronous BCI by Li et al. [68], which was designed to control a robotic manipulator. In that paper, a classifier was trained on five different classes; namely, the rest state and four MI classes, which were left-hand, right-hand, foot and tongue MI. From these five classes, they obtained six commands through the following pairings: left-hand MI followed by idle to move left, right-hand MI followed by idle to move right, tongue MI followed by idle to move up, foot MI followed by idle to move down, left-hand MI followed by foot MI to move forward, and right-hand MI followed by foot MI to move backward. 

In their 2018 paper, Yu et al. [1] presented a similar sequential MI-based paradigm, but applied it to asynchronous control of a wheelchair. The system was based on a linear discriminant analysis (LDA) classifier that could classify left-hand MI, right-hand MI, and idle-state data. The user could generate four commands using these three classes by carrying out these sequential commands: left-hand MI then idle, right-hand MI then idle, left-hand MI then right-hand MI, and right-hand MI then left-hand MI. The sequentially executed commands were determined by performing template-matching postprocessing on the output of the classifier. 

Sequential-command paradigms have also been used in conjunction with facial-movement commands. In their 2021 paper, Banach et al. [24] obtained seven different commands to control wheelchair movement from endogenous brain signals obtained from two activities; namely, eyes open and eyes closed. Unlike the works of Yu et al. [1] and Li et al. [68], which required the user to issue two-step sequential commands, Banach et al.’s study used a three-step sequential command setup in a synchronous manner. Since each step required three seconds, each command took up to nine seconds to perform, introducing an impractical latency into the system. 

Although sequential paradigms are able to increase the number of commands and have been shown to be effective for both synchronous and asynchronous BCIs, there is no rigorous comparison between a sequential command paradigm and a standard BCI paradigm. For example, Yu et al. [1] obtained four commands from three mental activities—left- and right-hand MI and idle—using the sequential MI approach; however, this was not compared to using a standard four-class MI paradigm consisting of, for example, left-hand, right-hand, legs, and tongue MI. Thus, several open questions remain, such as whether this two-step command paradigm with three mental activities will have a lower accuracy than a similar one-step command paradigm due to its sequential nature; in other words, does this still result in a higher accuracy than the four-class MI paradigm, which is expected to have lower accuracy? If the sequential paradigm results in better accuracy, is this worth the trade-off of higher latency? Since multistep paradigms require subjects to remember compound commands that may not be intuitive, does the cognitive load of such paradigms make them less practical and attractive to users? Thus, the true value of a sequential paradigm as opposed to a single-step paradigm is still open. 

### 9.2. Finite-State Machines

In their 2018 paper, Yu et al. [1] used a finite-state machine to increase the number of commands that could be issued by the subject to control a wheelchair. The previous subsection discussed how a sequential command paradigm was used to increase the number of commands from three to four. The finite-state machine was then used to increase the number of commands that the subject could issue to six, thus enabling them to trigger commands to start, stop, turn left, turn right, accelerate, and decelerate. This was because different sequential commands were linked to different actions in the wheelchair, depending on the current state of the system. For example, if the wheelchair was at rest, the left-idle sequential command could be used to start the wheelchair moving at a low speed, while issuing the same command while the wheelchair was already moving would result in an acceleration. A success rate of 94% was obtained for navigation tasks, which involved traversing a set route that included passing by specific waypoints and evading an obstacle. The “success rate” referred to the percentage of tasks successfully completed, with success being defined as reaching all the waypoints and not colliding with the obstacle. Together, the finite-state machine and sequential paradigm were effective in circumventing the issue of poor classification of the multiclass MI data by restricting the classifier to three classes. However, the finite-state machine required the user to remember the multiple-state-dependent meanings for some commands, which may have further increased the mental load on the user. 

### 9.3. Hybrid BCIs: Increasing the Degrees of Freedom through Additional Biosignals 

In this review, the term hybrid BCI (hBCI) refers to systems that use other biosignals in addition to EEG for driving an external device. Hybrid BCIs have primarily been used to increase the number of commands that can be issued through an EEG-based BCI by enabling the subject to issue commands through another biosignal paradigm [29,34,76]. 

In 2016, Soekadar et al. [34] presented an hBCI for controlling a motorized hand exoskeleton that combined EEG and electrooculogram (EOG) signals. EEG signals related to grasping intention were used to activate the exoskeleton to grasp. EOG signals related to horizontal eye movements were used to open the exoskeleton. This device was successfully used by subjects with quadriplegia and could be used to aid daily living activities based on grasping motions, such as holding a mug or book. 

Hybrid BCIs have been widely applied to BCI-controlled quadcopters [26,28,29,30], possibly because these devices present a large number of degrees of freedom and can be challenging to control even with standard manual controllers. Additional biosignals have been used primarily to increase the number of control commands, but they have also been used to increase system robustness [26,28,29,30]. In 2014, Kim et al. [26] presented a BCI-controlled quadcopter that used EEG and eye-gaze tracking. A camera on the hull of the quadcopter fed back a real-time image based on the quadcopter’s location, and the subject controlled the movement of the quadcopter using eye movements. There were two main control modes: in mode A, the subject could control the forward, backward, left translational, and right translational motion of the device by looking up, down, left, or right, respectively; using similar eye movements in mode B, the subject could control upward motion, downward motion, left rotation, and right rotation. Dual-modality commands were used to control take-off/landing (simultaneous concentration and eyes closed) and change between control modes A and B (concentration and looking at the center of the screen). 

Chen et al. [29] presented a BCI-controlled quadcopter that used left-hand and right-hand MI to control landing and take-off, and EOG signals to control navigation during flight. To control the direction of quadcopter movement (left, right, forward, or backward), the subject looked at a four-icon menu in which each icon was linked to a particular function. The hBCI was found to be effective, exhibiting online command decoding accuracies between 95% and 96% depending on the subject. However, restricting the user to focusing on a menu during flight limited this system’s practicality: when flying a quadcopter, the subject will usually be looking at the device, or in the least observing a livestream of its movements on a screen. 

Khan et al. [28] presented a more practical hBCI for controlling a quadcopter. The system could accept up to eight commands, four of which were supplied through EEG and four of which were supplied through functional near-infrared spectroscopy (fNIRS). The EEG-based commands, which used a facial-movement paradigm, were horizontal eye movements for landing, double blink to decrease altitude, eye movements in the vertical plane for anticlockwise rotation, and triple blink to move forward. The fNIRS commands consisted of mental activities, including imagined object rotation for take-off, mental arithmetic to increase altitude, word generation for clockwise rotation, and mental counting for backward movements. As a safety feature, the authors ensured complementary commands were assigned to different recording modalities. For example, the take-off command was issued using an EEG-derived command, whereas the “landing” command was issued using fNIRS. This ensured that if an erroneous command was interpreted using one modality—possibly due to a malfunction or poor signals being received at that time—a corrective command could be issued by the subject using the other modality. In the offline analysis, the fNIRS classifier obtained an accuracy of 77% and the EEG classifier obtained an accuracy of 86%. The paper did not provide a performance measure for the online flying task. Unlike the approach proposed by Chen [29], the subject could continuously observe the motion of the quadcopter during use. However, the commands lacked intuition, and forward movement required continuous triple blinking, which could induce fatigue. Furthermore, both works were tested in simple flying conditions that did not require any obstacle avoidance. 

### 9.4. Menu Navigation with Limited Commands

Some systems allowed subjects to choose the behavior of the dynamic device through a GUI-based menu that could be navigated using a limited number of commands. 

The menu used in the work of Gandhi et al. [67] sequentially presented subjects with pairs of options. The subject could pick one command from the five available in the menu to control the movement of a robot; namely, forward, backward, turn left, turn right, and halt. The menu also had a final option called “main” to return to the top of the menu. The options in the menu were organized in pairs, and the subject could select using left-hand or right-hand MI depending on whether they wanted to issue the command on the left or right, respectively. If the subject did not want to select either of the options, they could remain in the relaxed (idle) state, and the system would then show the next pair of commands. This enabled the subjects to select one of five commands using just two different MI activities and the rest state. 

Zhang et al. [71] presented a menu that could be used by the subject to select one of 25 different target points within a room to which they want the wheelchair to move. The menu was presented to the subject as a horizontal list of the numbers 1–25. The list was divided in two, and the user could choose to select either the left-hand side of the list by issuing a left-hand MI command or the right-hand side of the list by issuing a right-hand MI command. The selected half of the list was then displayed to the user with the halfway point indicated, and the user again selected the half of the list in which the desired location could be found. Eventually, the user was able to select the desired location from the final two destinations on the list. In all, the selection process had to be carried out five times in order to obtain the final destination, which resulted in a significant latency when using the system.

The menu-based approaches discussed in this subsection provided different levels of control to the user. Zhang et al. [71] proposed a semiautonomous system in which the subject could only select the desired destination, and then the wheelchair navigated autonomously to it. The implementation by Gandhi et al. [67] offered the subject lower-level control of the system by enabling them to select a dynamic action such as turning left. 

The approaches discussed in this section made use of the relatively high classification accuracy that can be obtained when just two MI classes are used. Nevertheless, the sequential selection process involved in these menus can be long and tedious, introducing latencies that may make them impractical for a real-life BCI.

## 10. Error Handling

Error handling in online BCIs is a cutting-edge problem. As previously discussed in Section 8, state-of-the-art BCI decoders experience spurious misclassifications that can affect performance. In addition to these spurious misclassifications, mental-state changes in the user, such as a drop in concentration, could also affect the performance of the BCI classifier. These events result in erroneous behavior, which can be especially undesirable in dynamic devices, since this could result in incorrect navigation or potentially dangerous collisions. 

When humans perceive that an error has occurred, spikes known as ErrPs appear in EEG signals [77]. Recent research has investigated whether ErrPs can be used to assist in the control of BCIs [72,77]. 

Ehrlich et al. [77] designed an ErrP-based decoder to asynchronously identify ErrP signals in a human subject observing a robot. The subject was aware of the task that the robot was required to carry out, and thus when the robot committed an error, ErrPs were generated in the subjects’ EEG data. When the ErrP decoder identified an error, the robot automatically corrected its performance. This study indicated that ErrPs were a viable paradigm for intuitive human–robot coadaptation. 

In a ground-breaking study, Bhattacharyya et al. [72] paired MI and ErrPs to control a robotic arm. The robotic arm had specific datums in a 2D plane that needed to be reached, and subjects could issue commands first to rotate the arm through right-hand MI, and then to change its planar position using foot MI. In order to stop the rotation or the planar movement at the datums, an ErrP decoder continuously monitored the data for ErrPs, and upon detection stopped movement of the arm. Although this was a basic BCI with limited applications, it was the only example in the reviewed literature that illustrated the use of asynchronous ErrPs to control an external dynamic device in conjunction with another paradigm, such as MI. 

## 11. Signal-Processing and Classification Techniques at the Cutting Edge

BCIs that are used for dynamic device control must be driven by reliable and high-accuracy EEG classification methods if they are to be used in practical applications. Although the requirement for high classification accuracy is not unique to BCIs for dynamic device control, since these devices often have to navigate complex environments or carry out fine motor tasks, precise control is even more crucial. Many of the studies in the reviewed literature focused on presenting novel classification techniques with the aim of improving classification performance. A high classification accuracy means that the user can obtain finer control of the dynamic system, enabling them to perform more precise movements, such as selecting a target object from a cluttered space using a robotic arm or navigating a sharp corner using a wheelchair. 

This section provides an overview of the signal-processing and classification techniques used, with a specific focus on novel signal-processing techniques that were presented for use in BCIs for dynamic device control.

The techniques used can be broadly categorized as either traditional machine-learning-based or deep-learning-based. Figure 6 gives an overview of this categorization. It is evident that traditional machine learning tended to dominate the literature, with deep learning methods, which are an emerging field when compared to traditional machine learning, constituting just 6% of the methods used (four papers). One paper merged both deep learning and traditional machine learning techniques to form ensemble classification techniques, and these accounted for the remaining 2%. The rest of this section discusses the traditional machine learning techniques used, followed by the deep learning techniques. 

### 11.1. Features for Traditional Machine Learning Techniques

Figure 7 and Figure 8 summarize the features and classifiers used in studies based on traditional machine learning techniques, and capture how frequently each method was used. Note that some studies used multiple features or multiple classifiers. In total, 61 studies used just traditional machine learning. 

When considering the results shown in Figure 7, it is immediately evident that common spatial pattern (CSP)-based features were the most popular. This was not surprising, since CSP features, which can be used to extract spatial features from EEG time series, have a long history as robust features in MI EEG classification [78], and 80% of the reviewed studies used an MI-based paradigm. Many studies used traditional CSP features [29,37,43,44,51,53,54,55,63,66], which were established for MI EEG classification [78]. However, filter bank CSP features, which have been shown to perform better than traditional CSP features [79], were also common [35,46,47]. Filter bank CSP feature extraction involves first passing the EEG signal through a filter bank, then extracting CSP features from each frequency band, and then performing feature selection to obtain the most discriminative subset of features [79]. Although this method improves the accuracy when compared to the traditional approach, the feature selection process can increase the training latency [80]. 

In total, 23 studies used time-domain modeling features (such as autoregressive features [48]) or multichannel time series data for classification [58], whereas 17 used frequency [17,72] or time-frequency domain features [65]. Time-series data are sometimes directly used for thresholding classification, which will be discussed in detail in the next subsection [58,81]. In real-time systems, low execution times are essential, and past research has indicated that time-domain features can have a similar performance to frequency-domain features, with the added benefit of a lower computational complexity [82]. 

Riemannian-geometry-based features are an emerging approach that was applied in two rehabilitative pedaling-machine BCIs [31,56]. They were used due to their low computational cost and strong classification performance [56]. 

Some studies used a mixture of different features in order to characterize EEG data. The aims of these approaches were to better represent MI EEG data, which exhibits spatial and frequency-domain changes between classes, and which evolves in time through event-related synchronization and desynchronization processes [83]. Many studies used CSP features to capture spatiotemporal characteristics in the data, and then paired them with time-domain features [37,45], time-frequency domain features such as the discrete wavelet transform [40,84], and/or functional brain network features [54]. Other studies paired time-domain features with time-frequency domain features [60,72,85] or frequency domain features [61]. 

Studies comparing the effectiveness of various features for online control of dynamic physical devices, both from an algorithmic performance perspective and from a user experience perspective, may provide insight into which features tend to result in the best overall performance. Unfortunately, no such study was found in the reviewed literature. 

### 11.2. Classifiers for Traditional Machine Learning Techniques

Figure 8 presents a bar plot of the classifiers used in traditional machine learning systems. Specifically, the heights of the bars indicate how frequently each classifier was used, while the stacking colors show which features were paired with each classifier. In this section, the prevalence of different classifiers is discussed first, then the classifier–feature pairings are analyzed in more detail. Note that the aim of this plot was to analyze the different feature–classifier pairings; this means that in papers that used different types of features in the feature vector but only one classifier, the classifier was counted multiple times in the plot shown (i.e., once for each feature). For example, Bhattacharyya et al. [72] used both time-frequency domain features and time-domain features to construct a feature vector that was input to a support vector machine (SVM) classifier. In order to construct Figure 8, the SVM classifier from that study was counted twice: once when paired with the time-domain feature category, and once when paired with the time-domain category. 

Discriminant analysis classifiers were overwhelmingly the most popular classifiers used in the literature. Of the classifiers that fell into this category, 25 were LDA classifiers, while information [86], quadratic [73], and spectral regression discriminant analysis classifiers, which mitigate the curse of dimensionality problem in LDA [54], were the others used. 

Although popular, the LDA classifier was mostly applied to binary classification problems: in 17 out of 25 instances in which the LDA classifier was used, it was applied to a binary classification problem. This may have been due to the fact that nonlinear decision boundaries may be generally more suitable for multiclass classification [87]. However, none of the papers provided an extensive study into the effectiveness of LDA as a multiclass classification approach to dynamic device control. Although LDA is widely used, it can also provide a baseline performance that can be used for comparison. 

Although LDA may be preferable due to its low computational training time [28], SVM was also a popular classifier due to its reliability [35,40]. Many studies applied the SVM classifier to binary classification [29,60,63,66], while other papers used expansions of the SVM classifier to enable multiclass classification [40,52]. Zhuang et al. [52] presented a majority-voting-based approach for four-class MI classification. This involved training six SVM classifiers in a one-vs.-one approach for each possible pairing of MI commands. The final classification label was assigned based on the class that obtained the most votes from the six classifiers. Gao et al. [40] used a directed acyclic graph to apply the SVM classifier to a three-class MI problem that consisted of MI for ascending stairs, descending stairs, and floor walking. This approach introduced hierarchical classification, with the top layer classifier trained on ascend-vs.-descend MI and the two lower-level classifiers trained on data for floor walking vs. ascend and floor walking vs. descend. 

Fuzzy-logic-based classifiers are a promising alternative to the popular SVM and LDA classifiers [14,55]. Andreu-Perez et al. [14] conducted an online mobile robot control task to compare their multiclass novel fuzzy classification approach, which they called GT2 FS, to other approaches in the literature, including LDA and SVM classifiers. The task began with the robot moving in an empty space. An obstacle was then placed in the space, and the subject had to use left- or right-hand MI to turn the robot to the other side (i.e., right or left). The aim was to avoid the obstacle placed in the robot’s path. The GT2 FS system obtained a classification accuracy of 86% for a three-class classification problem, which was notably better than that of the LDA classifier at 58%, and of the radial basis function SVM classifier at 61%. These BCI classification accuracy results were obtained while executing the robot navigation task. Within the proposed framework, each MI class was described using a number of fuzzy rules. These rules were based on prototypes built from the EEG data in an unsupervised way using the fuzzy Gath–Geva approach, as well as a small amount of training data that was used to assign prototypes to certain MI classes. Andreu-Perez et al. [14] also tackled a key problem in BCI classification: due to the nonstationarity of EEG data, misclassifications can occur over time as the online data diverges in its statistical properties from the data used for training. They proposed an online adaptation framework that used unsupervised learning to update the classifier, reducing the domain shift between the training data and incoming online data. 

Gaussian [13] and Bayesian posterior-probability [45] classifiers can model the inherent uncertainty in EEG data, producing probabilities associated with each class that give a reliable estimate of the certainty of the classifier [13]. These probability labels have been used to smooth the output BCI signal to prevent spurious misclassifications, and were discussed in more depth in Section 8.2.

Thresholding-based classifiers were used for paradigms that had highly distinctive EEG patterns [24,58]. Blinks, for example, could be easily identified as relatively large-amplitude spikes in the signal, and an amplitude threshold on the time-series data could be used to identify them [58]. A thresholding approach for power spectral density features was established to distinguish between the eyes-open and eyes-closed states, which was possible due to the notable alpha-band power that occurred when subjects closed their eyes [24]. Thresholding has also been used to discriminate between MI classes [38,42]. 

Relatively simple, shallow neural networks have been shown to be effective for endogenous device control [16,25,68,86]. Junwei et al. [25] presented a BCI-controlled wheelchair that was tested with subjects who had a neurodegenerative illness, and obtained a 93% success rate for navigation tasks in an everyday environment. The paradigm was based on mentally spelling the commands “LEFT”, “RIGHT”, “BACKWARD” and “FORWARD” to control the wheelchair accordingly. Two EEG channels, T3 and T4 were used. For feature extraction, a Chebychev filter bank with 22 frequency bands was used to filter the EEG data, and the band power for each band was extracted and summed across the channels used. The log transform of these values was then used as the feature vector. For classification, these features were input to a radial basis function neural network with a single hidden layer. Although this system had a high success rate, it operated in a synchronous way, which involved recording 5 s of EEG data, during which the subject issued the next command, executing the command and then repeating the process. This introduced an operational latency. 

Li et al. [59] presented an asynchronous control approach that used a multilayer feed-forward neural network and wavelet transform features to control movements of a prosthetic hand. Control was based on a facial-expression paradigm consisting of the expressions raised brow, furrowed brow, left smirk, and right smirk to issue the commands open hand, close hand, rotate wrist right, and rotate wrist left, respectively. In an online experiment that mimicked the hand movements required for drinking a glass of water, a success rate of 81% was obtained across subjects. 

Although effective, the works of Junwei et al. [25] and Li et al. [59] may not be ideal for practical BCIs. In the case of the work by Junwei et al. [59], the synchronous operation introduced approximately 7 s of latency between commands being issued. In the case of Li et al. [59], although the facial-expression paradigm enabled fine control of the prosthetic hand, this paradigm was not intuitive for a practical prosthetic. 

Ensemble learning of traditional machine learning classifiers has been shown to improve classification performance [33]. Kucukyildiz et al. [33] merged SVM, random forest (RF), and artificial neural network (ANN) classifiers for improved control of the direction of a BCI-controlled wheelchair. A hierarchical classification approach was used: if the certainty in the label assigned by the ANN was above a certain threshold, then the label assigned by the ANN was used. Otherwise, if the SVM and RF classifiers agreed on the classification label, this was assigned. If this was not the case, the individual output of the SVM classifier, then that of the RF classifier, were considered. If neither of the labels were assigned with certainties above a certain threshold, then the sample remained unclassified. This meant that the system did not take action on the data and continues operation in the previous control mode. Using the ensemble resulted in an average sensitivity of 85% across the three classes, compared to 65%, 76%, and 79% for the ANN, SVM, and RF classifiers alone, respectively. Unlike other studies discussed, this paper did not quote accuracy or success rate as measures of performance, so it could not be directly compared to other methods; however, this result did indicate that an ensemble of classifiers may be beneficial. 

Spychala et al. [37] proposed an ensemble learning technique based on logistic regression for a MI BCI to control a robotic hand. For the ensemble, they trained three logistic regression classifiers: one for distinguishing between the flexion MI and extension MI, one for distinguishing between the flexion MI and the idle state, and one for distinguishing the extension MI from the idle state. A finite-state machine for hand operation was constructed, and this determined which classifiers were consulted at different times of operation. Finally, probability thresholds for each classifier were tuned for each subject, and depending on whether the classifier probability outputs fell above or below these thresholds, different actions of the robotic hand were carried out. This approach was effective for online control of a BCI by stroke patients, obtaining a median online accuracy of over 60%. There was no comparison in the paper of the ensemble classifier to multiclass classifiers, so this is an opening for future work. 

The works of Kucukyildiz et al. [33] and Spychala et al. [37] illustrated that ensemble learning can be effective for dynamic device control. Studies into offline EEG classification have proposed a variety of novel ensemble learning techniques that could, in the future, be applied to online dynamic device control. Salient ensemble learning approaches applied to offline data have aimed to produce more generalizable classifiers that are robust to the artifacts and nonstationarities present in EEG data [88,89,90]. In 2019, Raza et al. [88] presented an adaptive ensemble learning technique. Covariate shifts in the incoming EEG data were identified by monitoring the CSPs in the data using an exponentially weighted moving-average filter. When shifts were detected, the ensemble of classifiers was expanded by adding new classifiers that were adapted in an unsupervised way to compensate for the shift. Later, Zuo et al. [90] proposed a cluster-decomposing-based ensemble learning framework. Clustering decomposition was used to divide EEG data into subsets that had different distributions, and each subset was used to train a different classifier. Then, multiobjective optimization was used to select the best set of classifiers. The proposed approach outperformed various benchmarking techniques. In 2022, Zheng et al. [89] proposed an ensemble learning technique based on temporal features, spatial features, and a multiscale filter bank. First, bootstrap sampling was used to divide the EEG data into subsets. From each subset, 20 different SVM classifiers were derived. These classifiers were obtained by preprocessing each subset using four different TDF techniques, thus obtaining four different versions of the subset. For each version, spatial, temporal, and TFD features were obtained. Then, five SVM classifiers were trained on different feature combinations, with some selected using feature selection and others not. Thus, five SVM classifiers were obtained from each of the four TFD-decomposed versions of the data. Weighted-decision fusion amongst the classifiers was then carried out using weights derived with the L2-norm method. The proposed approach outperformed other classification techniques applied to the same dataset. 

Linear mappings are an alternative classification approach [38,42,48,57]. In the mapping designed by Kobayashi et al. [57], time-domain features were extracted from the EEG data and then input to a linear formula that mapped the features onto class labels. In the approach presented by Xu et al. [48], a linear mapping was used to control the movements of a robotic arm in a 2D plane: the energy in the alpha and beta bands was estimated using 16th-order autoregressive coefficients, then weights were used to map these coefficients onto the position in the x–y plane that the arm was directed to move toward. 

CSP features were mostly used with SVM or LDA classifiers [35,47]. This type of feature–classifier pairing is a classic BCI signal-processing pipeline for MI EEG [43,50,91]. Time-domain features appeared to be the most versatile in the reviewed literature, being used with eight different classifiers [25,28,48,54,57,60,72,92]. Frequency domain features, time series, and CSP features have all been used with four different kinds of classifiers, indicating that they were also versatile. The results indicated that Riemannian, time-frequency domain, and functional brain network features may be emerging features, and future work could investigate how they perform with various other classifiers for dynamic device control. 

### 11.3. Deep-Learning-Based Techniques

In the simplest terms, deep neural networks are typically multilayer kernel networks with more than one hidden layer. However, deep neural networks are also characterized by novel architectural characteristics that have endowed them with various computational benefits. Very commonly, though not exclusively, the raw data is provided to the deep neural network, and then multiple neural layers effectively extract useful features for the task at hand. 

Due to the successful application of deep learning techniques in the domains of signal processing and computer vision, research interest in applying them to EEG data has been growing [41]. In recent years, research has shown that deep learning was especially effective for the processing of nonstationary and nonlinear data [2], confirming that they were appropriate for brain imaging and brain-signal decoding [41]. Novel deep learning networks such as the spiking neural network (SNN) are of particular interest to BCI researchers, since they are neuromorphic, meaning that their functionality imitates brain activity [12]. This kind of functionality is not available in traditional machine learning classification approaches. 

Deep learning systems have been used to facilitate asynchronous control that used more intuitive paradigms. In 2020, Tang et al. [2] presented a 1D convolutional neural network (CNN) approach for controlling a wheelchair. The multichannel EEG time series was arranged such that the electrodes on the left-hand side were grouped together first, followed by those on the right-hand side, followed by signals representing the difference between the signals on corresponding opposite sides of the scalp. In the CNN, EEG signals were processed along the time axis. The CNN had a three-branch architecture consisting of a small-scale kernel in the first branch, a large-scale kernel in the second, and a max-pooling layer for subsampling followed by a convolutional layer in the third branch. The features extracted from each branch were concatenated and then input to a classification layer. In online tests, accuracy ranged from 70% to 92%, depending on the command issued; the forward command, which was related to a facial-movement command, obtained the highest accuracy. In an offline analysis of a two-class classification problem for left- and right-hand MI, the 1D CNN approach obtained an average accuracy of 83%, which was better than the accuracies obtained for CSP and SVM-based classification (67%), a deep belief network (77%), and a CNN with a sparse autoencoder (78%) for the same experimental framework. However, the performance of the 1D CNN was not compared to state-of-the-art 2D CNN systems such as EEGNet [93] or ShallowConvNet [94]. 

In the same year, Jeong et al. [41] proposed a CNN and bidirectional long short-term memory (Bi-LSTM) network to control the movements of a robotic arm in 3D space. In this system, subjects imagined movements in the same arm in one of six directions; namely, up, down, left, right, forward, or backward. The movement of the arm was governed by a velocity profile output from the deep learning system that provided information about the velocities in the x, y, and z directions. Three 2D CNNs, which took as input the segments of EEG time-series data, were used to obtain velocity profiles for the x, y, and z directions separately. Afterward, the profiles were input to an LSTM network that determined the final velocity profile of the arm. Two online tasks were carried out to assess the performance of the system: picking up an object and drinking from a glass. Success rates for the first task ranged from 47% to 87%, whereas for the second task, they ranged from 33% to 57%. In an offline analysis, the performance of the CNN Bi-LSTM network was compared to state-of-the-art approaches using the normalized root-mean-square error (NRMSE) in the velocity profiles. The CNN Bi-LSTM had an NRMSE between 0.1150 and 0.2112, whereas EEGNET [93] had an NMRSE between 0.1412 and 0.3692. CNN Bi-LSTM was also found to perform better than ShallowConvNet [94] and an LSTM network. This approach provided the highest degree of freedom for a pure EEG paradigm of all the papers reviewed. However, a drawback of the system was that it did not facilitate the idle state and assumed that the subject was continuously issuing commands. 

In 2021, Batres-Mendoza et al. [12] proposed a SNN that used quaternion features for controlling a mobile robot [12]. Quaternions model EEG data in an abstract way in terms of rotations and orientations [12]. The mobile robot could turn left or right or continue forward based on left-hand MI, right-hand MI, or the idle state, respectively. This was the only deep-learning-based approach that used shared control: an ultrasonic sensor was used to detect obstacles, and if an obstacle was detected, the robot stopped and awaited commands to avoid the obstacle. While CNN networks aim to mimic the processing that occurs within the visual cortex [95], SNNs mimic the general biological behavior of neurons in the brain, which communicate via spikes in electrical activity [96]. The application of SNNs to BCIs is relatively novel, and it is promising for modeling the spatial and spectral characteristics of brain activities [96]. Batres-Mendoza et al. [12] did not directly compare the performance of the SNN-based system to CNN-based systems, which are state-of-the-art, through experimentation, so it was difficult to conclude whether SNNs provided any benefit. However, Batres-Mendoza et al. [12] did compare their results to those reported in other papers that used similar experimental conditions, and the proposed approach was the only one that obtained over 95% accuracy in online tests. Furthermore, the CNN-based classification approach had a response time that was between 0.1 s and 1.1 s shorter than those of the comparison systems, which consisted of CSP-LDA, CSP-SVM, and a hierarchical model–multilayer perceptron classifier.

The results in this section indicated that deep learning techniques have the potential to outperform traditional machine learning methods in terms of both classification accuracy and response times. Further research is required to identify whether deep learning systems always outperform traditional machine learning techniques, and if not, in which applications they provide the most benefit. 

### 11.4. Merging Traditional Machine Learning and Deep Learning Techniques

One study used ensemble learning to combine the outputs of machine learning and deep learning techniques [52]. Zhuang et al. [52] presented a BCI for driving a mobile robot based on four MI commands that corresponded to turn left, turn right, accelerate, and decelerate. The ensemble comprised a generic CNN classifier network that took EEG time-series data as input and a SVM classifier that took CSP features as input. The Adaboost.M1 boosting algorithm was used to merge the classification results. An offline study indicated that the ensemble was more effective than the individual classifiers, obtaining an average classification accuracy of 92%, compared to 87% for the CNN and 83% for the SVM classifier. Furthermore, this study used subject-independent training: the classifiers were trained using data from three subjects, then tested in the online experiment on another subject. Although the system performed well, further validation with more test subjects would be needed to conclude whether subject-independent training is most effective, or if possibly transfer learning—accomplished by pretraining the network on data from other subjects—should be used. 

These results illustrated that traditional machine learning and deep learning techniques can be complimentary, and ensembles that merge their capabilities can lead to overall improved performance. 

## 12. Subjects

Figure 9 is a histogram showing ranges for the number of subjects included in the reviewed studies, while Table 3 summarizes the number of patients involved in each study. The skew toward lower numbers of participants was expected, since recruiting subjects can be challenging. The study with the highest number of subjects (32) was a clinical trial [51] that involved significant resources and extensive participant sourcing. A total of 18% of the studies (12 papers) included subjects who had some relevant pathology that affected mobility. Of these studies, six recruited stroke patients [37,46,47,51,56,70], four recruited subjects with paraplegia or tetraplegia [15,19,34,75], one recruited spinal injury patients [69], and one recruited subjects with neurodegenerative diseases [25]. Although the recruitment of subjects with illness or disability can introduce additional challenges for researchers, it is an important aspect of BCI research, since many of these technologies are aimed at these individuals. Historically, BCI technologies have a low uptake among this population [3], and more extensive testing of BCIs with subjects who have significant disease or disability will help identify the best paradigms and control strategies for these stakeholders [3]. 

## 13. User-Experience Surveys

Only four of the reviewed papers reported formal results related to user experience with the BCI. Liu et al. [45] used standard questionnaires for subjective workload assessment; namely, the NASA Task Load Index (NASA-TLX) and the Subjective Workload Assessment Technique (SWAT). The NASA-TLX survey assesses the workload involved in operating a human–machine interface [97]. It factors in the mental, physical, and temporal demands of the system, as well as the effort required, the frustration experienced by the user, and the perceived performance of the system [97]. The SWAT method involves questions related to three areas of user experience; namely, time load, mental effort, and physiological stress [98]. For each question, subjects can rate the load they experience as low, medium, or high [98]. Results from these questionnaires were used to assess user experience when controlling a lower-limb exoskeleton using MI and to compare two different signal-processing algorithms for control: one based on sensorimotor rhythms (SMRs) and the other based on movement-related cortical potentials (MRCPs) [45]. SMRs are the classic activity that is observed over the motor cortex during MI, characterized by decreases and increases in alpha and beta band power. SMRs were generated by imagining left- and right-hand movements. MRCPs, which are signals associated with “readiness”, occur when planning or carrying out movements. In this study, they were a readiness potential that preceded the MI tasks being executed. Subjects reported that generating SMR data involved a higher workload than MRCP data. Furthermore, subjects who had a relatively poor performance with the SMR decoder reported more frustration when generating the data compared to MRCPs. Although MRCPs were favored by subjects, the average online accuracy obtained with the MRCP method was 69%, compared to 80% for the SMR method. These results indicated that the paradigm favored by subjects may not necessarily be the one that performs best. This result may have been due to MRCPs being relatively quick and easy to generate, since they only require the user to mentally express an intention to move, whereas the generation of SMRs requires concentration and imagination. 

Leeb et al. [5] also used a NASA-TLX survey to assess the change in load on the subject when controlling a wheelchair’s direction of travel using a BCI as opposed to a joystick. They noted that using the BCI did result in an increased mental load on the subject; however, in terms of frustration, perceived performance, and perceived temporal load, there was no significant difference between the two modes of operation for the subjects. 

Although standardized questionnaires provide an established assessment method that can be used for comparison across studies, they are generic. Some studies used specifically designed questionnaires that enabled the questions to be tailored to the device being assessed and the particular tasks being carried out in the experiments [37,51]. Spychala et al. [37] assessed subjects’ sense of ownership, agency, and binding when a robotic hand was used for neurofeedback during MI-based stroke rehabilitation. At a group level, stroke patients reported successfully feeling a sense of ownership, agency, and binding, indicating that this BCI modality could hold potential in a clinical setting. A clinical trial into another stroke rehabilitation system [51] that involved controlling a robotic glove or haptic robot through MI also used a specially designed questionnaire. The questions tackled physiological and comfort issues that may occur in use, perception of the system, and the usefulness of the system. Subjects reported finding the system tiring and uncomfortable. 

These studies illustrated how user feedback can provide valuable insight into the practicality of a BCI system. Despite this, the vast majority of papers reviewed did not record any data related to user experience. Although popular performance metrics such as accuracy, success rate, and information-transfer rate are vital assessment tools, if strong performance in terms of these metrics comes at the cost of an unacceptable cognitive load or a frustrating, tiring experience for the user, it is unlikely that these BCIs would be successful with a wide population of users. By collecting information about subjective user experience, researchers will be able to identify the paradigms and control schemes that subjects prefer, as well as forward-facing issues that need to be remedied. Thus, quantitative and qualitative approaches to user experience with BCIs need to be more widely adopted by researchers if these technologies are to become clinically and commercially viable products that are widely adopted. 

## 14. Conclusions: Emerging Questions and Future Work

A core issue across various systems is a lack of intuitive or practical commands. Some examples of these commands include using frowns and smirks to control the movement of a prosthetic hand [59], carrying out tongue MI to reverse a mobile robot [54], and continuously spelling out the word “FORWARD” to keep a wheelchair moving forward [25]. In particular, facial-movement paradigms have been touted as a promising complimentary or alternative paradigm to MI due to their high detection rate. Although this paradigm has been effective, it can be a source of impractical commands and should be applied with care. An example of an impractical command might be imagining tongue movement to increase the speed of a mobile vehicle—there would be no intuitive relationship between the command and the mental action. Furthermore, facial-expression-based systems are not a replacement for endogenous systems based on mental activity alone, which have the potential to be highly intuitive and provide seamless control. 

The feature extraction and classification techniques used had a fundamental effect on the performance of the BCI device. For use of endogenous systems based on mental activities to obtain a high level of functionality, more studies need to compare various classifiers and feature-extraction techniques in online systems. As previously discussed, it was common for studies to present novel signal-processing and classification approaches in isolation without any direct comparison to alternative techniques. Without such comparisons, the full value of the proposed approaches are still in question. 

Although offline analysis can provide important indications of the most suitable classifiers and features to use, online BCIs must be adequately robust to the domain shift and nonstationarities that can occur over time during BCI use. When using online BCIs, subjects may experience distractions, drops in concentration, and wandering thoughts, which can all impact the performance of the system. Thus, it is recommended that researchers not only rely on classifiers and features that have worked well in offline studies, but also seek to investigate and address these issues that are unique to online BCI use. This was seen in some of the literature reviewed, such as in the work of Andreu-Perez et al. [14], who proposed an unsupervised method for updating a fuzzy-logic-based classifier.

Overwhelmingly, systems in the literature depended on the MI-based paradigms. In fact, 80% of the papers reviewed included MI as a paradigm. Although this is an established and reliable paradigm for users who are able to generate MI signals, the MI systems suffered from a high BCI illiteracy rate and a low number of degrees of freedom [28,29]. One way of addressing this is through hybridization of the BCI. However, hBCIs that are used to increase the number of commands available in the BCI usually have impractical setups, such as BCIs for flying quadcopters that require the subject to either focus on a GUI [29] or issue commands by carrying out mental arithmetic [28]. More research is required into hBCI systems that are based on intuitive control paradigms and that seamlessly integrate different biosignals for enhanced control without increased impracticality. Sequential commands and finite-state machines have also been used to expand the control capabilities of MI-based BCIs [1], but these may increase the mental load on the user, particularly in the early user-training stages. 

Alternative paradigms to MI could be explored in future works. A synchronous spelling paradigm was shown to be effective for device control by persons with neurodegenerative illnesses [25]. The next step would be testing this paradigm in an asynchronous system. Furthermore, speech imagery, which involves imagining speech to issue BCI commands, showed promising performance in a GUI-based application [99]. In future works, speech-imagery paradigms could be used to control external dynamic devices with simple but effective commands. 

ErrPs can be used for asynchronous error correction in BCIs. Bhattacharyya et al. [72] showed how ErrP signals can be detected asynchronously and used to stop the movement of a robotic arm. This study also illustrated that ErrP signals could be used in conjunction with a MI paradigm to control an external device. However, the experiments carried out were simplistic, with the robotic arm just needing to pass specific predefined datums. The next step may involve investigating the effect of ErrP-based correction in devices performing more complex tasks, such as an MI-controlled robot or a wheelchair navigating an everyday living environment. 

Sophisticated shared control is also required for navigation in complex environments. Many systems in the literature used basic shared control that led to an emergency stop, leaving the user to carry out obstacle avoidance, which could be mentally fatiguing [12,24,33]. Ideally, BCIs for dynamic device control have integrated obstacle avoidance. Although some approaches in the literature proposed more sophisticated approaches, they were still limited to maximum margin avoidance [52] or selection of a target object in an uncluttered environment [69]. In practical systems, subjects may want to pass near “obstacles”; for example, they may want their wheelchair to pass near a table to pick something up, a task for which obstacle avoidance algorithms that maximize the distance from obstacles would be inappropriate. Potential fields have been used to successfully enable subjects to navigate close to objects that may otherwise be classified as obstacles; however, potential fields have a known issue: when traveling down corridors, both walls can be detected as obstacles, and this can cause the path of the vehicle to oscillate. Moreover, the potential-fields approach can become locked in local minima, leading to the device they are controlling becoming trapped and being unable to move [100]. Furthermore, subjects can have different risk tolerances, and can have different BCI literacies. Shared-control systems that can be tuned to the preferences and capabilities of subjects represent a new research frontier in the control of dynamic devices through BCIs. 

Finally, user-experience surveys can provide insight into the practicality and intuition of BCIs [37,45,51]. More studies should use quantitative and qualitative techniques to assess user experience, since this can provide invaluable insight and avenues for future works. The low uptake of BCIs amongst some target users has been linked to a lack of knowledge of the needs and expectations of subjects who experience disability or diseases that affect mobility [3]. Thus, it is especially important that studies involving such subjects prioritize the gathering of feedback on user experience, in addition to quantitative performance measurement. 

## Figures and Tables

**Figure 1 sensors-22-05802-f001:**
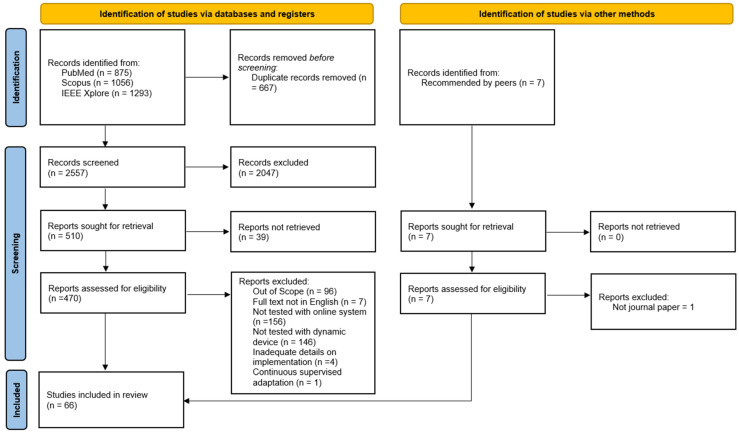
A summary of the paper selection process formatted using the template available at [6].

**Figure 2 sensors-22-05802-f002:**
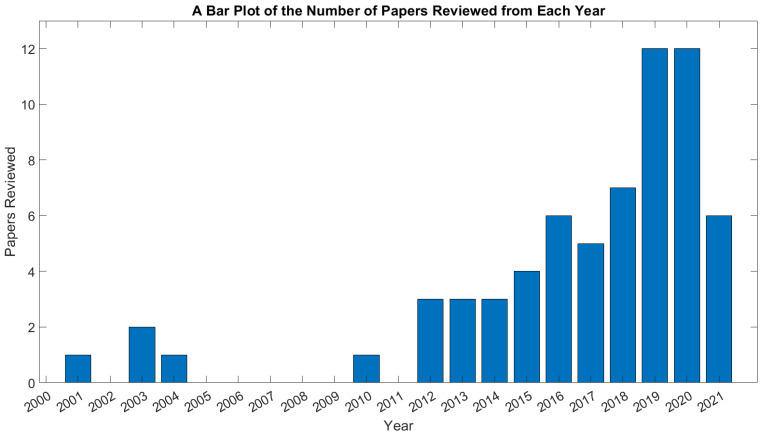
A year-by-year breakdown of the reviewed literature.

**Figure 3 sensors-22-05802-f003:**
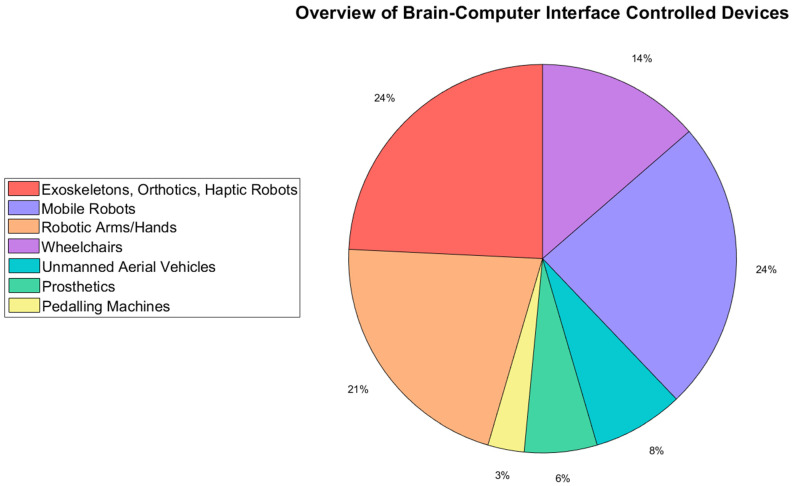
A pie chart showing a breakdown of all the different BCI-controlled devices in the literature reviewed.

**Figure 4 sensors-22-05802-f004:**
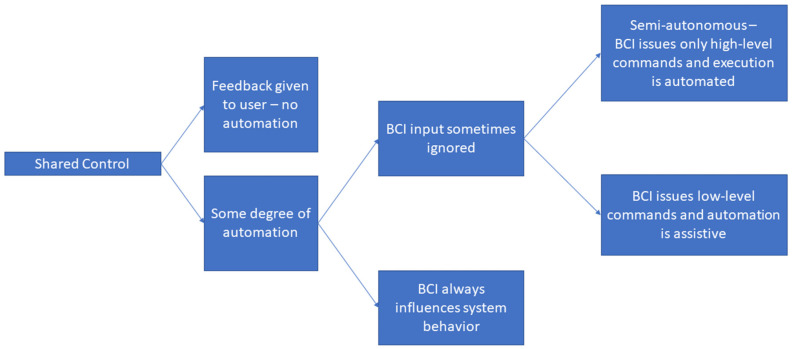
A taxonomy of the shared-control approaches proposed in the reviewed literature.

**Figure 5 sensors-22-05802-f005:**
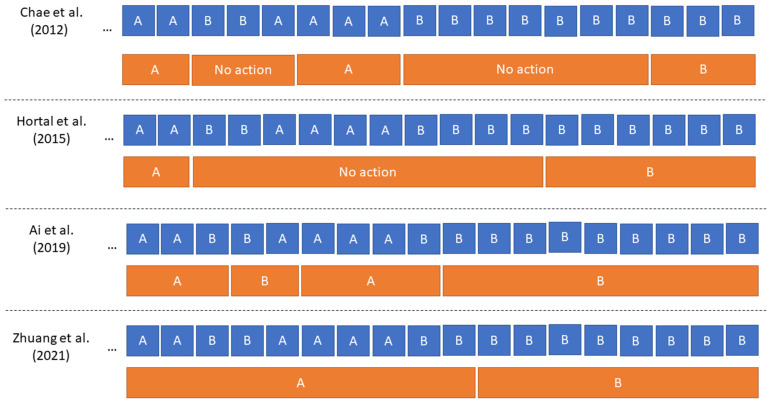
Comparing different false-alarm approaches. The blue bars show the BCI classifier output at the previous time step, and the orange bars show the decision made at the current time step. The example is for a two-class problem in which A denotes the classifier label for one mental state and B is the label for the other state. Each mental state was related to a different movement in the dynamic device. During “no action” phases, movement of the device was paused. In this example, it was assumed that at the start, the BCI classifier was outputting in class A for a long period (more than eight consecutive samples). Four different approaches are presented, namely those by Chae et al. [73], Hortal et al. [74], Ai et al. [54] and Zhuang et al. [52].

**Figure 6 sensors-22-05802-f006:**
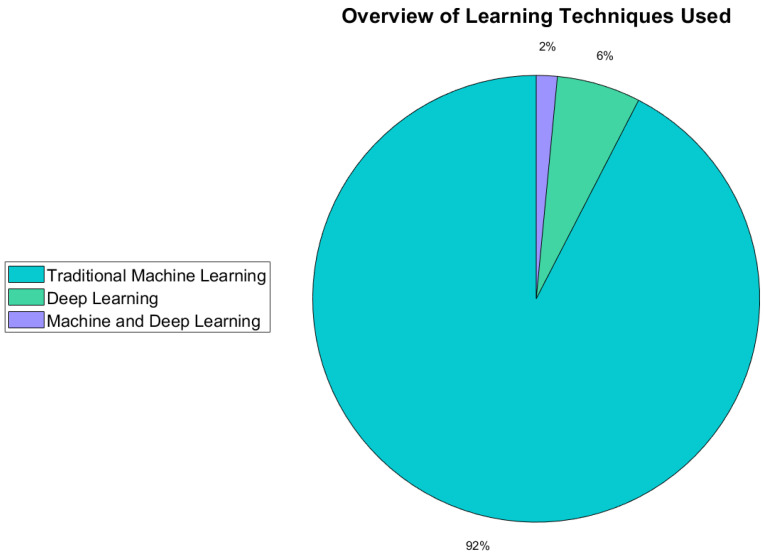
A pie chart illustrating the proportion of studies that used traditional machine learning and deep learning techniques.

**Figure 7 sensors-22-05802-f007:**
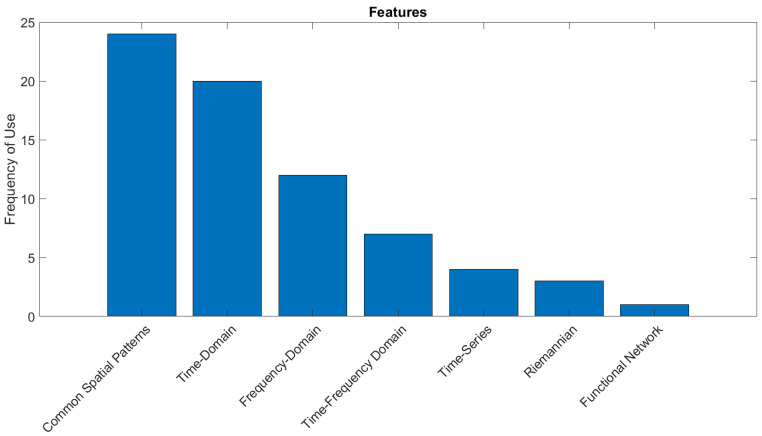
A bar plot showing the features used in machine learning systems.

**Figure 8 sensors-22-05802-f008:**
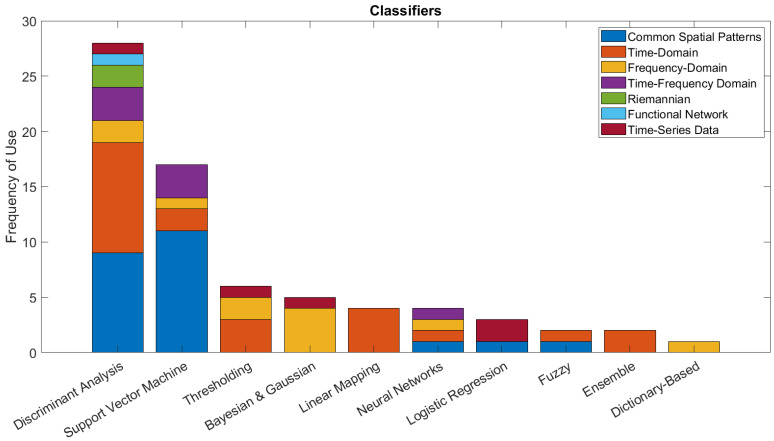
A bar plot showing the classifiers used in machine learning systems with indications of the features used with each classifier group.

**Figure 9 sensors-22-05802-f009:**
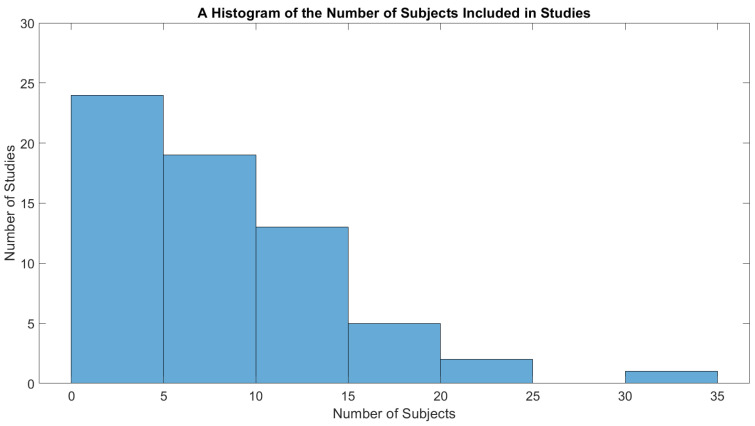
A histogram showing ranges of the number of subjects included in the studies reviewed. For each range, the right-hand number was included and the left-hand number was excluded.

**Table 3 sensors-22-05802-t003:** A summary of the papers that included patients and the number of subjects considered.

Paper	Condition	Number of Subjects
Spychala, 2020 [37]	Stroke	7
Romero-Laiseca, 2020 [56]		2
Moldoveanu, 2019 [51]		32
Cantillo-Negrete, 2018 [47]		6
Ang, 2018 [46]		9
Frisoli, 2012 [70]		4
Soekadar, 2016 [34]		6
Do, 2013 [75]	Paraplegia or tetraplegia	10
Pfurscheller, 2003 [19]		1
Pfurscheller, 2001 [15]		1
Kim, 2019 [69]	Spinal injury	2
Junewi, 2019 [25]	Neurodegenerative disease	4

## Data Availability

Not applicable.

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
