# Peer review of "A Comprehensive Review of Endogenous EEG-Based BCIs for Dynamic Device Control"

_sensors, 2022, doi:10.3390/s22155802_

Round 1
Reviewer 1 Report
This paper is a review of using endogenous EEG based brain computer interface (BCI) to control dynamic devices. Experimental paradigms, control strategies, the type of devices, decoding techniques, as well as users’ experiences are covered in the manuscript. Overall this manuscript is well written, but some improvements are still expected. Below are my comments:
Section 6, Although published studies in this area is enormous and cannot be all covered by a single review paper, there are still some important work you need to include in the manuscript.
For BCI based robotics control:
Edelman BJ, Meng J, Suma D, Zurn C, Nagarajan E, Baxter BS, Cline CC, He BJ. Noninvasive neuroimaging enhances continuous neural tracking for robotic device control. Science robotics. 2019 Jun 19;4(31):eaaw6844.
Meng, J., Zhang, S., Bekyo, A. et al. Noninvasive Electroencephalogram Based Control of a Robotic Arm for Reach and Grasp Tasks. Sci Rep 6, 38565 (2016).
For BCI based exoskeleton control:
Soekadar SR, Witkowski M, Gómez C, Opisso E, Medina J, Cortese M, Cempini M, Carrozza MC, Cohen LG, Birbaumer N, Vitiello N. Hybrid EEG/EOG-based brain/neural hand exoskeleton restores fully independent daily living activities after quadriplegia. Science Robotics. 2016 Dec 6;1(1):eaag3296.
Section 6, There is one more type you need to add, which uses EEG based BCI to activate patients’ own paralyzed muscles via functional electrical stimulation, like the one reported in this paper:
Müller-Putz, Gernot, and Rüdiger Rupp. "The EEG-controlled MoreGrasp grasp neuroprosthesis for individuals with high spinal cord injury–multipad electrodes for screening and closed-loop grasp pattern control." International Functional Electrical Stimulation Society 21st Annual Conference. 2017.
Section 9.3, For BCI with hybrid signals, one of the aforementioned studies should definitely be included, given its importance in this area:
Soekadar SR, Witkowski M, Gómez C, Opisso E, Medina J, Cortese M, Cempini M, Carrozza MC, Cohen LG, Birbaumer N, Vitiello N. Hybrid EEG/EOG-based brain/neural hand exoskeleton restores fully independent daily living activities after quadriplegia. Science Robotics. 2016 Dec 6;1(1):eaag3296.
Page 23, Line 803, LDA based methods are widely presented in many studies, but sometimes they only act as a baseline to be compared with.
Page 24, Line 813, “Fuzzy classifiers” is not appropriate, should be “fuzzy logic based classifiers”.
Page 26, Line 913, “Deep Learning based Techniques” seems to be a better title for this section.
Sections 11.2, 11.3, 11.4, I don’t think “deep learning” and “machine learning” are two mutually exclusive concepts. “Machine learning” should be a superset containing both “traditional machine learning” and “deep learning”. To be more accurate, you intended to review “traditional machine learning” based classifiers in 11.2, while “deep leaning” based classifiers in 11.3. You can keep all these sections, don’t need to merge them, but need to revise some words/sentences to make sure the concepts are all right.
Section 11.3, Probably need to summarize why deep learning based methods were proposed for EEG recognition. What are the particular problems for EEG-BCI that deep learning can solve but traditional machine learning cannot? Do deep learning based methods always overcome traditional machine learning based methods? If not, in what kind of scenarios deep learning is preferred?
Section 11.4, The aim of ensemble learning is not just “merging machine learning and deep learning techniques”. First, “machine learning” and “deep learning” are not two mutually exclusive concepts, as I pointed out above. Second, ensemble learning uses multiple learning algorithms together to achieve better performances, but these algorithms don’t need to be deep learning based. Ensemble learning has been widely applied in BCI studies, but Ref [41] is the only one mentioned in the manuscript (probably because it performed actual device control). You may need to add more refs for this section. For endogenous EEG classification tasks, here are a few refs you can review and include in this section (only about algorithms, may not perform external device control):
Zuo C, Jin J, Xu R, Wu L, Liu C, Miao Y, Wang X. Cluster decomposing and multi-objective optimization based-ensemble learning framework for motor imagery-based brain–computer interfaces. Journal of neural engineering. 2021 Mar 2;18(2):026018.
Raza H, Rathee D, Zhou SM, Cecotti H, Prasad G. Covariate shift estimation based adaptive ensemble learning for handling non-stationarity in motor imagery related EEG-based brain-computer interface. Neurocomputing. 2019 May 28;343:154-66.
Zheng L, Feng W, Ma Y, Lian P, Xiao Y, Yi Z, Wu X. Ensemble learning method based on temporal, spatial features with multi-scale filter banks for motor imagery EEG classification. Biomedical Signal Processing and Control. 2022 Jul 1;76:103634.
Page 30, Line 1092, Again, change the word “fuzzy classifier”.
Reviewer 2 Report
1- The paper is well-written and required minor corrections, for example ‘’M1’’ which should be MI. Also, I suggest to double check the abbreviations which are introduced.
2- Due to large number of abbreviations, I suggest the authors to prepare a section named ‘‘NOMENCLATURE’’ regarding the journal format.
3- Authors claimed that it is a review paper from on different BCI applications, whilst a recent published review paper (year 2021) on the BCI for controlling mobile vehicles, aerial vehicles and bionic (humanoid) robot arms is not presented. Also, I found several papers from 2019 to 2021 in this review paper that are not presented or referenced. It is suggested to consider the following reference:
Hekmatmanesh A, Nardelli PH, Handroos H. Review of the state-of-the-art of brain-controlled vehicles. IEEE Access. 2021 Jul 27.
In page 22, the CSP feature and SVM classifier and Hybrid-BCI introduced. It is suggested to consider the usage of different modifications of the CSP and SVM in the abovementioned reference. Also, several Hybrid-BCI papers are available
4- It would be interesting to prepare a column in Tables 1 and 2 to present the accuracies.
5- In the literature review, the advantages and disadvantages of the methods and employed EEG-based patterns are explained, but a few of methods are not presented. Also, I would like to suggest that please explain the source of limitation comes from the employed pattern or the algorithm.
Round 2
Reviewer 2 Report
The comments are well answered and I am satisfied from the comments.